# Identification of a super-functional Tfh-like subpopulation in murine lupus by pattern perception

Stefanie Gryzik[1†]*, Yen Hoang[1,2†]*, Timo Lischke[1], Elodie Mohr[1]*, Melanie Venzke[1], Isabelle Kadner[1,2], Josephine Poetzsch[1,2], Detlef Groth[2], Andreas Radbruch[1,3], Andreas Hutloff[1], Ria Baumgrass[1,2]*

[1]German Rheumatism Research Center (DRFZ), A Leibniz Institute, Berlin, Germany; [2]University of Potsdam, Potsdam, Germany; [3]Charité, Campus Mitte, Berlin, Germany

*For correspondence:
s_gryzik@web.de (SG);
yen.hoang@drfz.de (YH);
elodie.mohr@drfz.de (EM);
baumgrass@drfz.de (RB)

†These authors contributed equally to this work

Competing interests: The authors declare that no competing interests exist.

**Abstract** Dysregulated cytokine expression by T cells plays a pivotal role in the pathogenesis of autoimmune diseases. However, the identification of the corresponding pathogenic subpopulations is a challenge, since a distinction between physiological variation and a new quality in the expression of protein markers requires combinatorial evaluation. Here, we were able to identify a super-functional follicular helper T cell (Tfh)-like subpopulation in lupus-prone NZBxW mice with our binning approach "pattern recognition of immune cells (PRI)". PRI uncovered a subpopulation of IL-21$^+$ IFN-$\gamma^{high}$ PD-1$^{low}$ CD40L$^{high}$ CXCR5$^-$ Bcl-6$^-$ T cells specifically expanded in diseased mice. In addition, these cells express high levels of TNF-$\alpha$ and IL-2, and provide B cell help for IgG production in an IL-21 and CD40L dependent manner. This super-functional T cell subset might be a superior driver of autoimmune processes due to a polyfunctional and high cytokine expression combined with Tfh-like properties.

## Introduction

CD4 T cells play a major role in autoimmune diseases such as systemic lupus erythematosus (SLE) by accumulating as autoreactive disease-promoting memory cells and amplifying inflammation as well as providing B cell help (*Suárez-Fueyo et al., 2016*). It is known that dysregulated cytokine production, metabolic alterations and aberrant cell signalling cause abnormalities in T cell differentiation and hyperactivation of B cells and thus contribute to lupus disease (*Moulton et al., 2017*). For SLE, there are several studies in mouse models and humans, showing that the progressive disease is associated with lower expression of TNF-$\alpha$, IFN-$\gamma$, and IL-2 (*Humrich et al., 2010*) as well as higher expression of PD-1 (*Jiao et al., 2014*), IL-21 (*Wang et al., 2014*), and IL-10 (*Facciotti et al., 2016*). Despite the detection of these individual parameters, a comprehensive combinatorial characterization of T-cell subsets in SLE is still lacking.

Recent articles have revealed enormous and unexpected heterogeneity and complexity in T cell subpopulations with partially overlapping properties and functions of the cell subsets, such as follicular T helper (Tfh) and Tfh-like cell subsets (*Trüb et al., 2017*; *Wong et al., 2015*; *Wong et al., 2016*). Of particular interest are the recently observed human T peripheral helper cells (Tph), which expand in autoimmune diseases (*Bocharnikov et al., 2019*; *Christophersen et al., 2019*; *Ekman et al., 2019*; *Rao et al., 2017*). These and other studies show on the one hand the enormous potential to gain new biological insights from multidimensional cytometry data. On the other hand, they reveal the tremendous demand for new analysis and visualization approaches, especially for the identification of small but different characteristics in apparently homogeneous subpopulations, which is emphasized by numerous review articles (*Kvistborg et al., 2015*; *Mair et al., 2016*; *Newell and*

*Cheng, 2016*; *Saeys et al., 2016*). Most of the current analysis strategies are based on clustering and the respective frequencies of the clusters (*Levine et al., 2015*; *Qiu et al., 2011*; *Spitzer et al., 2015*; *Wong et al., 2016*). The main problem with these methods is that they tend to find mainly cell-populationclusters which are significantly high in phenotypical contrast. If the cell subpopulations are very similar or even partially overlapping, as in the present investigation, then a cluster approach is usually not very meaningful (*Newell and Cheng, 2016*; *O'Neill et al., 2013*; *Saeys et al., 2016*).

To circumvent this problem, we developed the bin-based 'pattern recognition of immune cells (PRI)' strategy. PRI is a reproducible analysis and visualization approach to examine the combinatorial expression patterns of proteins in a mixture of similar cell subpopulations. Here, we have combined PRI with comprehensive cytometric measurements of Th cells from NZBxW F1 mice, a common lupus model. The results of our approach provide evidence for a super-functional Th cell subpopulation mainly characterized by IL-21$^+$ IFN-$\gamma^{hi}$ PD-1$^{low}$, TNF-$\alpha^{hi}$ IL-2$^{hi}$, and functional properties of Tfh-like cells.

## Results

### Introduction into visualization of multi-parametric flow cytometry data with pattern recognition of immune cells (PRI)

To analyze pathogenic memory Th cell subsets (CD4$^+$ CD44$^+$ cells (*Swain, 1994*), in the following called Tmem cells), we used the NZBxW mouse model, which is one of the best-established models for human SLE, and visualized the combinatorics of three proteins from splenic T cells of old diseased mice (chronic inflammation, high proteinuria). We plotted IFN-$\gamma$ and TNF-$\alpha$ expression of Tmem cells with conventional tools such as contour plots (*Figure 1A*), a pie chart (*Figure 1C*), a stacked bar chart (*Figure 1D*), and the recent tool 'color mapping of dots' by FlowJo (*Figure 1B*). As expected, IFN-$\gamma$ is almost exclusively produced by CD44$^+$ cells (*Figure 1D*).

We confirmed this fact with our PRI visualization tool by plotting different features of IFN-$\gamma$ expression as heat maps onto the x-y plane of TNF-$\alpha$ *vs.* CD44 (*Figure 1E*). To this end, the x-y plane was divided into equally sized bins. The resulting plots were called bin plots. Per bin, we calculated and depicted the features (i) cell density, (ii) frequency of IFN-$\gamma$-producing cells (%), (iii) mean fluorescence intensity (MFI) of IFN-$\gamma$, and (iv) the relative expression level of IFN-$\gamma$ calculated as mean fluorescence intensity of IFN-$\gamma$ only of IFN-$\gamma^+$ cells (MFI+ (IFN-$\gamma$)) (*Figure 1F*). Apparently, our bin plot comparison shows that the bins with the highest expression levels of IFN-$\gamma$ are TNF-$\alpha^{hi}$ CD44$^+$ (*Figure 1F*) and are in an area with low cell density and low-to-intermediate frequency of IFN-$\gamma$ (*Figure 1E*). Such information cannot be extracted from the conventional plots.

Next, we used PRI for 'pseudo-multi-parametric viewing' by color-coding different parameters of the same sample (old diseased NZBxW mouse) on the same x-y plane. This visualized that the IFN-$\gamma^{highest}$ bin area contains cells that are also high in IL-2, but low in PD-1 (*Figure 1F*). The applied pseudo-multi-parametric viewing of PRI supports the discovery and characterization of specific subpopulations with certain or unique properties not only in an easy to understand and comprehensive manner but also on a reproducible basis.

### Disease kinetics reveal a subsequent reduction of poly-functional memory T cells

Despite the suspected importance of combinatorial expression of inhibitory receptors (such as PD-1) and cytokines for immunity (*McKinney et al., 2015*), to our knowledge, there are no comprehensive co-expression studies with CD4$^+$ T cells at single-cell protein level for SLE.

To characterize the disease kinetics of lupus development, we scored NZBxW mice according to their age and proteinuria (*Figure 2A*), determined their auto-antibody concentrations and analyzed the expression of PD-1 and four main cytokines in CD4$^+$ Tmem cells. Increased frequencies of PD-1$^+$ and IL-10$^+$ cells as well as decreased frequencies of IL-2$^+$ and TNF-$\alpha^+$ cells were age- and disease-dependent (*Figure 2A–C*). The co-production of the cytokines changed in the course of the disease to a lower degree of poly-functionality, visible by the accumulation of non-producing and IFN-$\gamma$ single-producing cells (*Figure 2—figure supplement 1*).

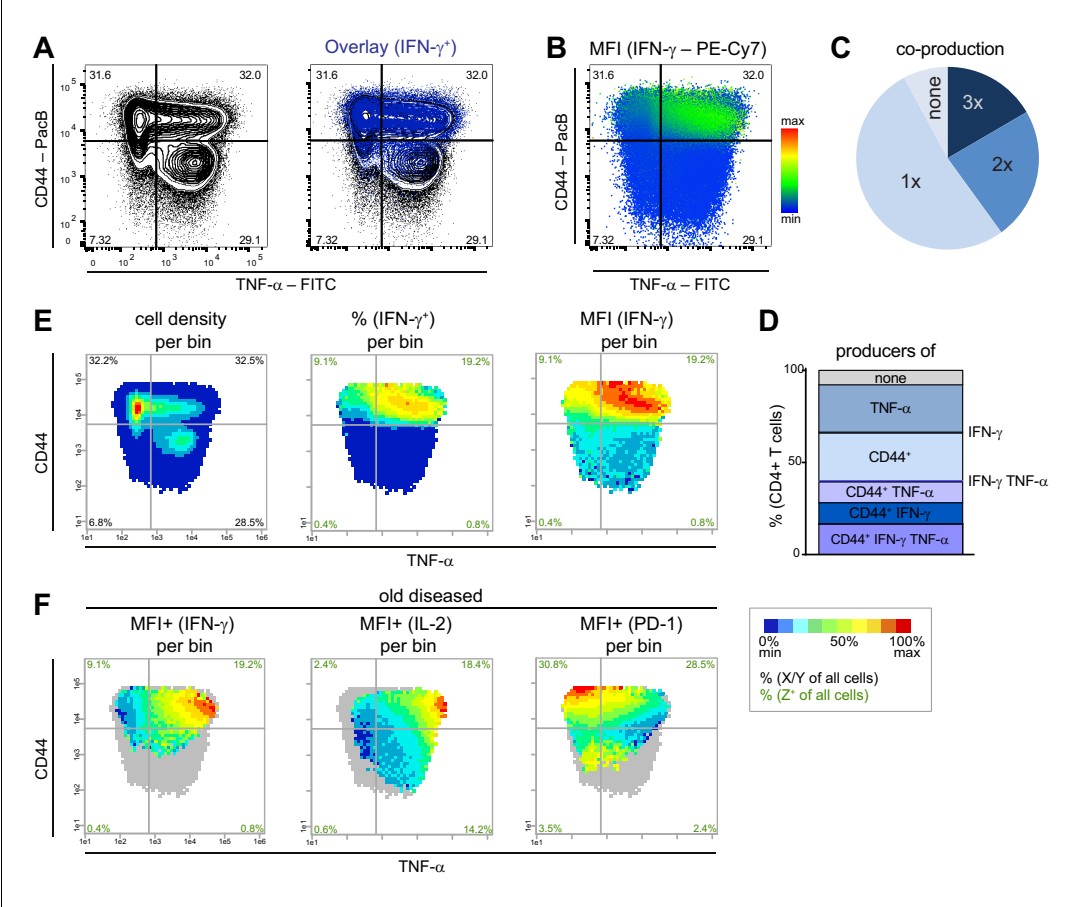

**Figure 1.** PRI allows the visualization of the combinatorial protein expression. Flow cytometry data from stimulated (PMA/ionomycin) splenic T cells from NZBxW mice were analyzed either conventionally or by bin plots. (A) Overlay of IFN-γ+ cells on all T cells. (B) FlowJo's color maps showing MFI of IFN-γ. (C, D) Frequencies of the co-production of TNF-α, CD44+ and IFN-γ are depicted as categories (C) and as individual combinations (D). (E) For bin plots, the x-y-plane of TNF-α and CD44 is divided into small, equally sized bins (asinh = 0.2). If a bin contains the minimum number of cells (10), a statistical feature is calculated such as cell density (left), frequency (middle) or MFI (right) of IFN-γ. The scale of the feature is represented as a pseudo-color code. (F) Relative mean expression level per bin (MFI+) of IFN-γ only in IFN-γ producing cells (left), of IL-2 in IL-2+ (middle) and PD-1 in PD-1+ cells (right). Grey bins contain less than 10 Z+ cells. Data represent one old diseased mouse (A–F). (E, F) Cell frequencies per quadrant are calculated from the number of cells per sample (black) and number of Z+ cells per sample (green). Data represent three independent experiments with (C, E) n = 4 mice per group.

The online version of this article includes the following source data for figure 1:

**Source data 1.** *Figure 1C,D*: Frequencies of double and single producers.

As loss of poly-functionality is considered as a hallmark of chronic infections and often used as an end-point to evaluate exhaustion of CD8+ and CD4+ T cells (*Larsen et al., 2012*; *Tilstra et al., 2018*), we investigated the combinatorial expression of the four cytokines with PD-1.

## PRI facilitates the combinatorial characterization of chronically activated T cells

We classified the disease scores 1–2 as 'young', and 3–5 as 'old diseased' and compared these scores using combinatorial analyses. According to publications on chronic infections (*Hwang et al., 2016*), we divided the PD-1 expression into three ranges (negative (-), low to intermediate (low), and high (hi) (*Figure 2C*), using FMO controls and loss of function in IL-2 expression and plotted the diagrams for conventional analysis (*Figure 2D and E*) and PRI visualization (*Figure 2F*). During disease progression of lupus nephritis, a higher expression of inhibitory receptor PD-1 was associated with a lower degree of CD4+ Tmem-cell functionality regarding IL-2, TNF-α and IFN-γ expression (*Figure 2D—figure supplement 2A and B*). This was underlined when comparing PD-1+ (low and hi)

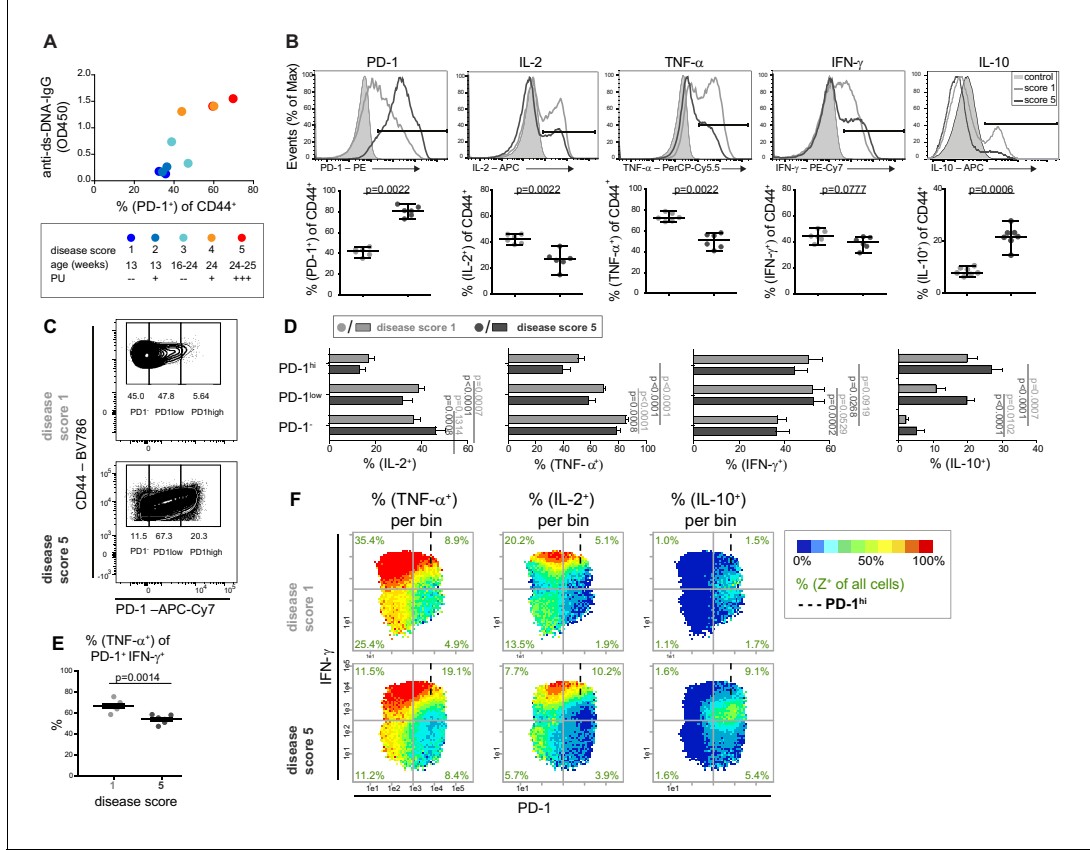

**Figure 2.** The cytokine co-expression is altered in PD-1+ T cells in young and old mice. (A) Disease status of NZBxW mice was scored according to their age and proteinuria (PU). The disease score was correlated with the frequencies of PD-1+ T cells and serum levels of anti-ds-DNA-IgG antibodies. (B) Disease-associated changes of the protein expression shown by histogram overlays (top) and statistical analyses (bottom). (C) Representative gating scheme of PD-1−, PD-1low and PD-1hi subpopulations in CD4+ CD44+ T cells from mice of disease score 1 and 5, respectively. (D) Statistical analyses of the frequencies of IL-2, TNF-α, IFN-γ and IL-10 producers in PD-1 subpopulations from mice with disease score of 1 (light grey) and 5 (dark grey). (E) Frequencies of TNF-α + cells of the PD-1+ IFN-γ+ subpopulation in disease score 1 and 5. (F) Representative bin plots of disease score 1 and 5 with PD-1 (x-axis), IFN-γ (y-axis) displaying the frequencies of TNF-α, IL-2 and IL-10, respectively, per bin. Cut-off for PD-1hi cells is marked with dashed lines. Data represent two independent experiments with (A), n = 2 mice for each score, (B– F) n = 6–7 mice per group. Samples were compared using the Mann Whitney test (B), a repeated measure two-way ANOVA with Geisser-Greenhouse correction and Dunnett's multiple comparison test (D) and a two-sided unpaired t-test (E). Data are presented as the mean ± SEM.

The online version of this article includes the following source data and figure supplement(s) for figure 2:

**Source data 1.** *Figure 2A*: Serum levels of anti-ds-DNA-IgG vs. PD-1 frequency.

**Source data 2.** *Figure 2B*: Frequencies of protein expression of mice in disease score 1 and 5, respectively.

**Source data 3.** *Figure 2D*: Frequencies of IL-2, TNF-α, IFN-γ and IL-10 producers in PD-1 subpopulations of mice in disease score 1 and 5, respectively.

**Source data 4.** *Figure 2E*: Frequencies of TNF-α cells of the PD-1+ IFN-γ+ subpopulation in mice of disease score 1 and 5, respectively.

**Figure supplement 1.** The combinatorics of the co-expression of IFN-γ, TNF-α and IL-2 in CD44+ T cells is altered with disease progression.

**Figure supplement 1—source data 1.** *Figure 2—figure supplement 1B*: Frequencies of boolean combinations of the co-expression of cytokines in mice of disease score 1 to 5.

**Figure supplement 2.** PD-1+ T cells exhibit differential cytokine expression levels compared to PD-1− T cells in young and old diseased mice.

**Figure supplement 2—source data 1.** *Figure 2—figure supplement 2B*: Mean fluorescence intensity in PD-1 subpopulations of mice in disease score 1 and 5, respectively.

**Figure supplement 2—source data 2.** *Figure 2—figure supplement 2E*: Frequencies of boolean combinations of the co-expression of IL-2, TNF-α, IFN-γ and IL-10 in PD-1 subpopulations in disease score 1 and 5, respectively.

**Figure supplement 3.** Bin plot patterns are reproducible and statistically robust.

**Figure supplement 3—source data 1.** *Figure 2—figure supplement 3D*: Frequencies shown in the upper right quadrant in the bin plots of mice in disease score 1 and 5, respectively.

and PD-1− cell populations in mice of score 5: the frequencies of IL-2- and TNF-α-producing cells

were lower but that of IFN-γ- and IL-10-producing cells were higher in the PD-1$^+$ cell populations (*Figure 2F—figure supplement 2B-D*). The frequencies of TNF-α$^+$ Tmem cells within the IFN-γ$^+$ PD-1$^+$ CD4$^+$ subpopulations were reduced with lupus nephritis (*Figure 2E*), similar to CD8$^+$ Tmem cells in chronic infections (*Utzschneider et al., 2016*).

In contrast to pie charts (*Figure 2—figure supplement 2E*), pseudo-multi-parametric viewing of bin plots allows to compare whole patterns (frequencies and expression level) and bin areas of different cytokines which are color-plotted on the same x-y-plane. Apparently, bins in the IFN-γ$^{hi}$PD-1$^{−/low}$ area contain the highest frequencies and expression levels of TNF-α and IL-2, but almost no IL-10 production (*Figure 2F—figure supplement 2C and D*). The low co-expression of IL-10 and IL-2 as well as TNF-α was confirmed by contour plots (*Figure 2—figure supplement 3A*). Compared to the conventional analysis of the PD-1 subgroups (*Figure 2D—figure supplement 2A, B and E*), the higher combinatorial complexity of PRI enabled the identification of more informative subpopulations like PD-1$^{low}$ IFN-γ$^{hi}$ cells (*Figure 2F—figure supplement 2C*).

Obviously, the combinatorial pattern of three parameters in bin plots (*Bendfeldt et al., 2012*) supports a more conclusive statement than just determining the frequencies of quadrants after subsequent gating, which demonstrates a particular strength of our semi-continuous bin plotting. This might explain the contrary results of Kasagi et al. which report 'higher levels of IFN-γ in PD-1$^{hi}$ than in PD-1$^{low}$ cells' in diseased NZBxW mice in the CD4$^+$ population (*Kasagi et al., 2010*).

## Reproducible and statistically robust bin plot patterns exemplarily shown with combinatorial IL-10 expression

We discovered a similar bin plot pattern of IL-10 (higher frequencies associated with a PD-1$^{hi}$ IFN-γ$^{low}$ phenotype) in all studied mice despite high differences in the frequencies of IL-10 expression per mouse and disease score (*Figure 2—figure supplement 3B and D*), and even when mice from a different experiment were concatenated (three mice each) (*Figure 2—figure supplement 3C*). Using IL-10 bin plots, we also demonstrated some general advantages of the PRI approach: (i) Varying the cut-off value for IL-10$^+$ cells had a high impact on IL-10 frequencies, but almost no influence on their bin pattern (*Figure 2—figure supplement 3E*). (ii) The plotted bin area with 10 cells per bin provides statistically reliable data, as demonstrated by the relative standard error of the mean (RSEM) (*Figure 2—figure supplement 3F*). (iii) A comparison of different samples regarding IL-10 expression, here different donor mice, different experiments and different disease scores, is easily feasible due to their bin pattern similarities.

## Bin plots revealed an IL-21$^+$ super-functional T cell subpopulation

Recently, IL-10 was found to be highly co-expressed with IL-21 (60%) in Tmem cells in SLE patients (*Facciotti et al., 2016*). Therefore, we additionally included the measurement of IL-21 in the combinatorial expression study of NZBxW T cells, which led to 32 possible combinations for the co-production of five cytokines (*Figure 3—figure supplement 1A*). Contrary to *Facciotti et al., 2016*, however, in NZBxW mice we observed neither substantial IL-10 and IL-21 co-expression with conventional contour plots nor strongly overlapping bin regions in bin plots on the TNF-α and IFN-γ plane (*Figure 3A and D*). In contrast, IL-21 is particularly co-expressed with IL-2 and TNF-α$^{hi}$ as demonstrated with contour plots and bin plots analyzing two (*Figure 3A*), three (*Figure 3D*) or four cytokines (*Figure 3E*) simultaneously.

When searching for IL-21$^+$ Tmem cell subpopulations with differences in frequencies between young (score 1–2) and old diseased (score 3–5) NZBxW mice, we identified a tendency towards higher frequencies of IL-21$^+$ and IL-21$^+$ IFN-γ$^+$ cells in old compared to young mice using conventional gating analysis. However, they were not statistically significant in contrast to PD1$^{hi}$ cells of Tmem and CD44$^+$ cells of CD4$^+$ subset (*Figure 3C*).

Gating for PD-1$^{hi}$ and PD-1$^{low}$ populations revealed that all populations studied had higher frequencies in older mice, but only the PD-1$^{low}$ IL-21$^+$ IFN-γ$^+$ population was significantly higher (*Figure 3C*). These data are comparable with the PRI-data (*Figure 3—figure supplement 1B*), even if the PD-1$^{low}$ IL-21$^+$ IFN-γ$^+$ population was not significantly different between young and old diseased mice. Although IL-21 is a hallmark of Tfh cells, in the NZBxW model, the expression of IL-21 is not correlated with a PD-1$^{hi}$ CXCR5$^+$ Tfh cell phenotype (*Figure 3G—figure supplement 1D*).

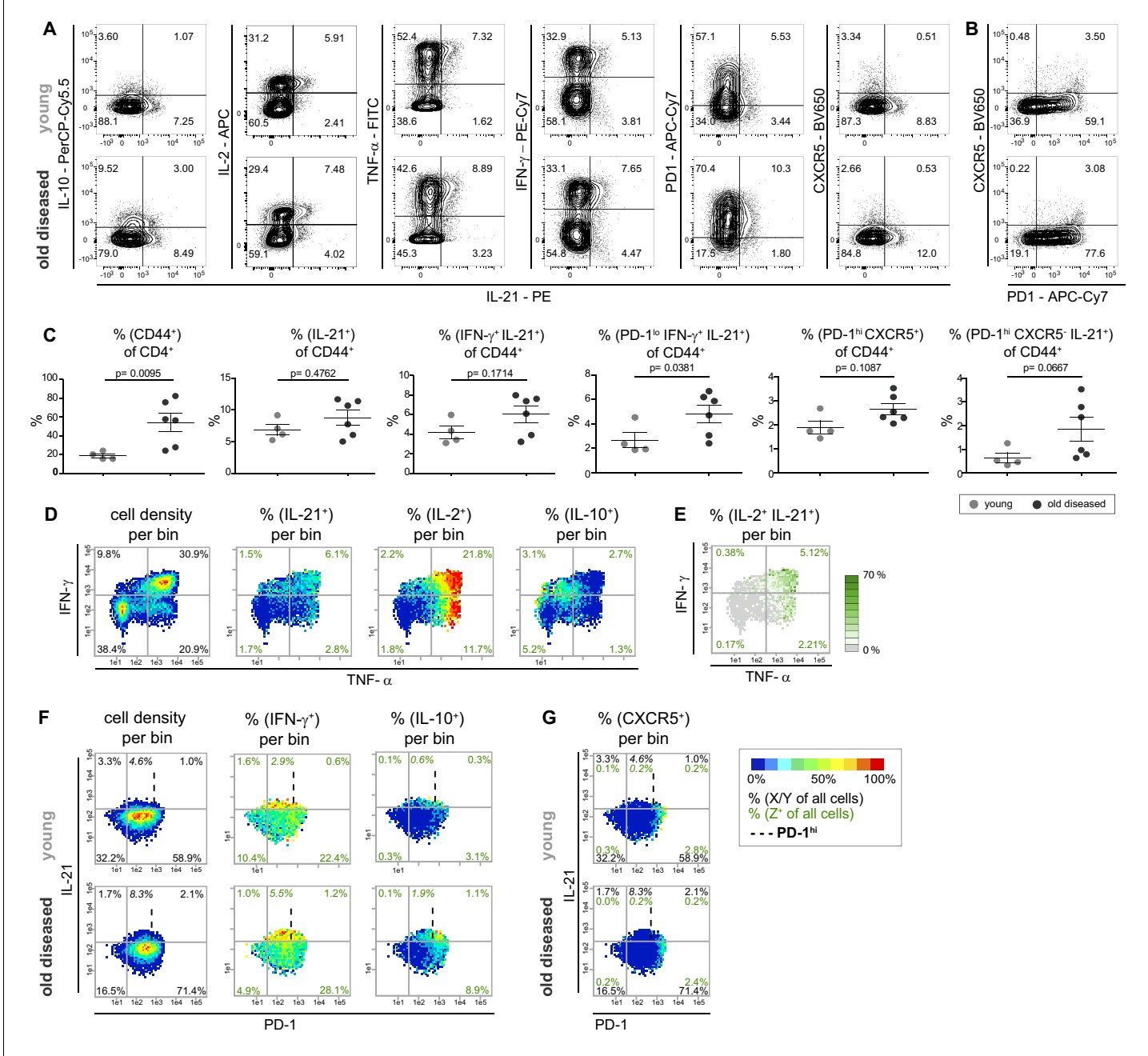

**Figure 3.** Most IL-21 producers in the NZBxW model strongly co-express Th1 cytokines but no CXCR5. (A) Co-production of IL-21 with cytokines and the Tfh markers PD-1 and CXCR5 in concatenated CD4+ CD44+ cells from young and old diseased mice. (B) Tfh cells were identified based on their high expression of PD-1 and CXCR5 positivity. (C) Statistical analyses of IL-21+ subpopulations extracted from FlowJo. (D, E) Cytokine co-production is analyzed by bin plots showing the frequencies of IL-21+, IL-2+, IL-10+ (D) and IL-2+ IL-21+ cells (E). Green numbers indicate the frequency of IL-2+ IL-21+ double producers relative to all cells per quadrant (E). (F, G) Bin plots of concatenated samples of young and old diseased mice with PD-1 (x-axis), IFN-γ (y-axis) displaying the cell density and frequencies of IFN-γ+ and IL-10+ (F) and CXCR5 (G), respectively, per bin. Percentages written in italic letters represent the population size of PD-1low IL-21+ IFN-γ+ cells. Data are representative of three independent experiments (A–B). (C) Pooled data from two experiments involving n = 4 (young) and n = 6 (old) mice per group. Samples were compared using Mann-Whitney-U-test. Data are presented as the mean ± s.e.m. (D–G) The samples were concatenated from n = 3 young and old mice of the same experiment. Cut-off for PD-1hi cells is marked with dashed lines. (E) If a bin contains the minimum number of cells (5), the frequency of the third marker is shown in pseudo-colors. Cell frequencies per quadrant are calculated on the number of cells per sample (black) and number of Z+ cells per sample (green).

The online version of this article includes the following source data and figure supplement(s) for figure 3:

**Source data 1.** *Figure 3C*: Proportion of different CD4+CD44+ T cell subsets in young score 1-diseased mice versus old score 5-diseased mice.

*Figure 3 continued on next page*

*Figure 3 continued*

**Figure supplement 1.** The majority of IL-21 is produced by non-Tfh cells.

**Figure supplement 1—source data 1.** *Figure 3—figure supplement 1A*: Raw data to determine the frequencies of boolean combinations of coexpression of five cytokines.

**Figure supplement 1—source data 2.** *Figure 3—figure supplement 1B, C*: Frequencies from IL-21$^+$ subpopulations extracted from PRI bin plots.

Obviously, the frequencies of IL-21$^+$ cells among Tmem cells do not change significantly with higher disease scores, but rather the properties of these cells to co-produce IFN-γ and PD-1 (*Figure 3C and F—figure supplement 1B and C*). The identified PD-1$^{low}$ IL-21$^+$ IFN-γ$^{hi}$ cell subpopulation even expressed high amounts of IL-2 and TNF-α in both young and old diseased mice. Therefore, in the following they are referred to as super-functional T (Tsh) cells.

## Pseudo-multi-parametric viewing of bin plots to analyze co-expression properties

Based on our experience (*Fuhrmann et al., 2016*) that the co-expression of cytokines and receptors matters, we further characterized the identified IL-21$^+$ IFN-γ$^{hi}$ PD-1$^{low}$ Tmem cells by studying the bin pattern of 42 functionally relevant proteins on the PD-1 and IFN-γ plane (17 out of them are shown in *Figure 4A*; *Figure 4—figure supplement 1A and B*). We used IFN-γ instead of IL-21 as the second parameter on the y-axis because the intensity range of IFN-γ is much wider than the range of IL-21 and therefore better for visualizing preferred expression areas of the most Z-parameters. Due to the reproducible expression patterns, PRI allowed the analysis of proteins from different mice and experiments. Again, PD-1$^{low}$ IFN-γ$^{hi}$ bins showed the highest probability of IL-21$^+$ cells. T cells in these bins predominantly co-expressed CXCR3 and T-bet and lacked FoxP3 expression. The IL-21$^+$ expression pattern overlapped only partially with that of Bcl6, CXCR5 or CXCR4 (*Figure 4A—figure supplement 1A*). Furthermore, the IL-21$^+$ IFN-γ$^{hi}$ PD-1$^{low}$ Tmem cells additionally produced CD40L and ICOS, two receptors involved in the interaction of T and B cells. CD40L was even produced in the highest amounts by IFN-γ$^{hi}$ cells (*Figure 4B—figure supplement 1A*). In contrast, the stimulatory receptors ICOS, OX40, GITR and CD27 showed lower intensities of expression in the PD-1$^{low}$ IFN-γ$^{hi}$ bin area but higher intensities in the Tfh (Bcl6$^{hi}$) bin area. The same is true for the inhibitory receptors TIGIT, CTLA4 and BTLA, which were mainly produced by Tfh cells, chronically activated T cells and/or Treg cells (*Figure 4B—figure supplement 1*). All these properties and their intermediate-to-high PSGL-1 expression suggest a Tfh-like cell subpopulation with Th1 properties instead of extrafollicular Tfh cells (*Choi et al., 2015*; *Odegard et al., 2008*).

As the IL-21$^+$ IFN-γ$^{hi}$ PD-1$^{low}$ Tmem cells lacked CXCR5 expression as other Tfh-like cells, we addressed their contribution to IL-21$^+$ cells (*Figure 5A and B*) in spleen and their tissue distribution (*Figure 5C*). We were able to localize them in spleen and non-lymphoid organs, liver and lungs at comparable frequencies (*Figure 5C and D*). We could also detect them in kidneys and peripheral blood (*Figure 5C and D* and data not shown). Again, IL-21$^+$ cells co-produced high levels of IFN-γ in all these organs, but almost no CXCR5 (*Figure 5D*). Interestingly, the Tsh cell subset in diseased NZBxW mice is two times bigger than the Tph and 10 times bigger than the Tfh cell subset.

## PRI reveals differences between Tfh and Tfh-like cell subpopulations

To better define and compare important properties of Th cell subpopulations, we determined expression values of bin areas with multiple samples. To this end, we covered the PD-1-IFN-γ plane with a 3 × 3 grid, which led to nine categories according to their PD-1 and IFN-γ expression levels (*Figure 6A*). The statistical analysis of many markers within these categories (*Figure 6B—figure supplement 1A*) confirms a particular pattern of Th cell subpopulations (shown as a schematic distribution in *Figure 6D*). These include the highest frequencies of IL-21$^+$ T cells in PD-1$^{low}$ T cells, which was demonstrated by their normalization to CD44$^+$ T cells (*Figure 6C*) and their positive correlation with the IFN-γ expression level (*Figure 6B*).

Using viSNE plots, the association of IL-21$^+$ cells with high IFN-γ expression and their separation from PD-1$^{hi}$ T cells could also be demonstrated (*Figure 6—figure supplement 1B*). However, viSNE was only partially appropriate for dim markers such as CXCR5 or Bcl6, because it illustrates the MFI over all cells. Color Maps (FlowJo 7.2) have similar issues with suboptimal color distributions for dim

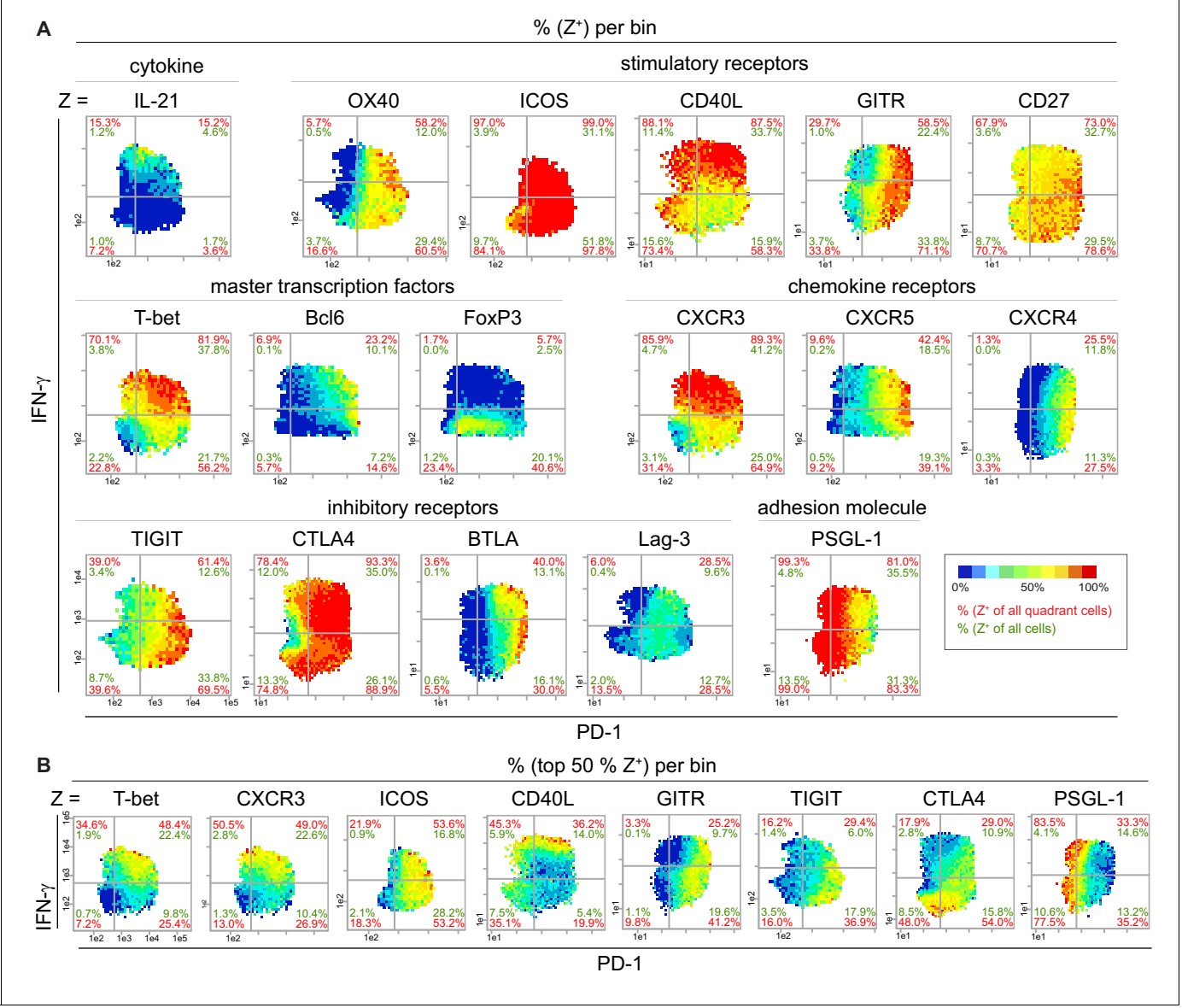

**Figure 4.** Super-functional T cells exhibit Th1 characteristics and are CD40L^hi ICOS^+. (**A**) Bin plots visualize the co-expression of PD-1 and IFN-γ with various proteins in stimulated (PMA/ionomycin) splenic T cells of old mice. (**B**) Distribution of the top 50% Z^+ cells of selected markers from (**A**). If a bin contains the minimum number of cells (10), the frequency of the third marker is shown in pseudo-colors. Cell frequencies per quadrant are calculated on the number of Z^+ cells in the quadrant (red) and number of Z^+ cells per sample (green). Data are representative for at least two independent experiments with n ≥ 3 mice.

The online version of this article includes the following figure supplement(s) for figure 4:

**Figure supplement 1.** Super-functional T cells are CD40L^hi ICOS^+.

markers and only represent MFIs and not expression levels (***Figure 6—figure supplement 1D***). PRI, instead, allows the calculation of relative expression levels of marker Z of Z^+ cells (MFI+ (Z)) (***Figure 6—figure supplement 1C***).

To further delineate Tfh-like and Tfh cell subpopulations, we investigated the co-expression of IL-21 with the Tfh marker proteins CXCR5 and Bcl6 as well as the T/B cell interaction receptors ICOS and CD40L. Again, we used PRI to calculate the frequencies of positive cells, this time for two defined marker proteins per bin to visualize double producers (***Figure 6E***). CXCR5^+ Bcl6^+ (Tfh cells) predominantly appeared in bins at the PD-1^hi side, while there were only few bins with IL-21^+ Bcl6^+, IL-21^+ CXCR5^+ and even IL-21^+ ICOS^hi T cells. Instead, the co-production of IL-21 and CD40L is

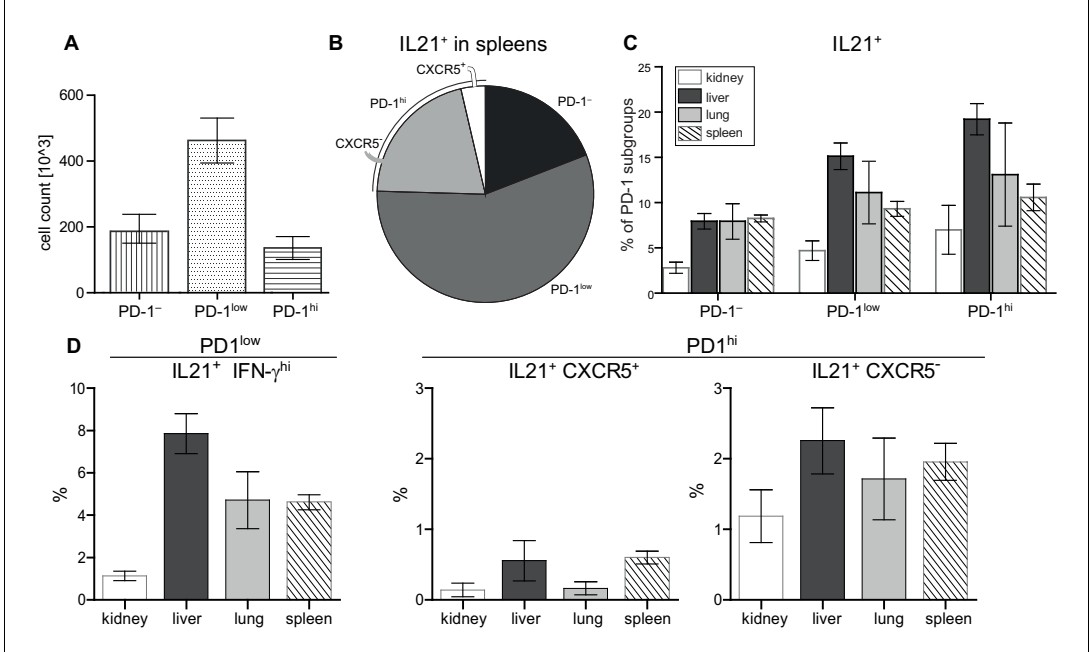

**Figure 5.** Super-functional T cells in peripheral organs exceed extrafollicular T cells and Tph cells in terms of frequency. (A) Absolute numbers of PD-1 subsets in spleens. (B) Frequency of IL-21 producers in spleens. (C, D) Frequency of IL-21 producers in terms of localization and in terms of PD-1 subset. Data represent two independent experiments with n = 4 mice per organ. Data are presented as the mean ± s.e.m.

The online version of this article includes the following source data for figure 5:

Source data 1. *Figure 5A*: Frequencies of PD-1 subpopulation.
Source data 2. *Figure 5B*: Frequencies of IL-21 producers in spleens.
Source data 3. *Figure 5C*: Frequencies of IL-21 producers in terms of localization and in terms of PD-1 subset.
Source data 4. *Figure 5D*: Frequencies of IL-21 producers in terms of localization and in terms of PD-1 subset.

localized in bins with IFN-γ$^{hi}$ T cells (*Figure 6E*). IL-21 expression levels did not correlate with Bcl6 and negatively with CXCR5 (*Figure 6G*).

Plotting only IL-21$^{+}$ CXCR5$^{-}$ cells in PRI-bins with a PD1$^{hi}$ threshold visualized Tsh and Tph cell subsets together. Most of these cells are not PD-1$^{hi}$ but PD-1$^{lo}$ (*Figure 6F*).

The discrepancy between IL-21 and Tfh marker expression became evident again when the frequencies (*Figure 6H*) or expression levels (*Figure 6—figure supplement 1E*) of IL-21 and Bcl6 producers were visualized by 3D surface plots, since positive cells for each protein appeared in different areas.

## Super-functional T cells provide B cell help

Next, we asked for similarities and differences between already described Tfh-like cell subpopulations (*Choi et al., 2015*; *Hutloff, 2018*; *Vu Van et al., 2016*) and the specified IL-21$^{+}$ IFN-γ$^{hi}$ PD-1$^{low}$ Tmem cells identified here. The main difference is the high PD-1 expression which has been described for all previously identified Tfh-like subsets, including the human Tph cells (*Bocharnikov et al., 2019*; *Christophersen et al., 2019*; *Ekman et al., 2019*; *Rao et al., 2017*).

Expression level, number and frequency of cytokines were usually not characterized in parallel with Tfh-like marker proteins. The most important similarities between the here specified and already known Tfh-like cells, besides IL-21 expression, are the absence of CXCR5 and expression of ICOS and CD40L$^{hi}$ (*Figure 4A and B—figure supplement 1*). These characteristics have prompted us to investigate whether the IL-21$^{+}$ Tmem cells are able to induce antibody production by co-culture of CD44$^{-}$ naive B cells and CD44$^{+}$ CXCR3$^{+}$ non-Tfh non-Treg cells (*Figure 7A*). CXCR3$^{+}$ served as a surrogate marker for IFN-γ production in FACS live sorting and helped to enrich IL-21$^{+}$ IFN-γ$^{hi}$ Tmem cells by a factor of five compared to CD44$^{+}$ T cells (*Figure 4A*, *Figure 7—figure supplement 1*). Polyclonal stimulation of these CXCR3$^{+}$ T cells induced a high IgG production by B cells

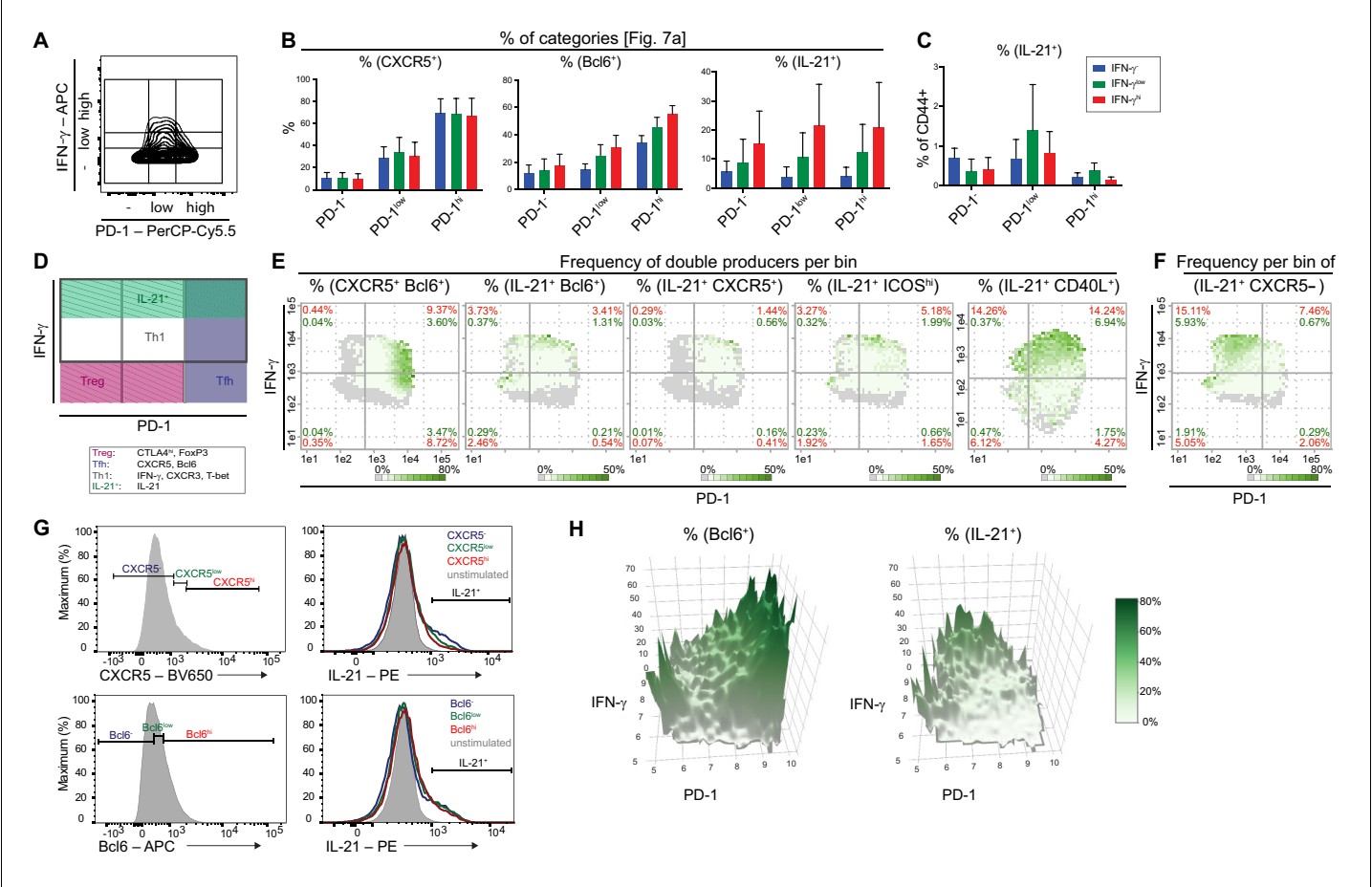

**Figure 6.** Comparison of Tfh and Tfh-like cell subpopulations by bin patterns. (A) Subdivision of Tmem cells into nine categories based on their PD-1 and IFN-γ expression (compare to D). (B) Barplots depicting the frequencies of CXCR5, Bcl6 and IL-21 producers in respective subpopulations. (C) Frequency of the IL-21 producers of each subpopulation relative to CD44+ cells. (D) Areas with highest probability for Treg, Tfh, Th1 and IL-21+ cells. (E) Representative bin plots of the frequencies of double producers for CXCR5/Bcl6, IL21/Bcl6, IL21/CXCR5, IL21/ICOS[hi], IL21/CD40L in splenic T cells of old diseased mice. (F) Representative bin plot of the frequency of IL21 producing, but CXCR5 non-producing cells in splenic T cells of old diseased mice. (G) T cells were sub-divided according to their CXCR5 and Bcl6 expression levels into negative (-), low and high (hi) expressors (left). Their IL-21 production was investigated by histogram overlays (right). (H) 3D heatmap showing frequencies of Bcl6 (left) and IL21 producers (right) with PD-1 (x-axis) and IFN-γ (y-axis). All data represent three independent experiments with (B, C), n = 3–9 mice and (E–G), 3–11 mice. E-F, Frequencies of 'double' producers are calculated per quadrant (red) and per all CD44+ cells (green). Grey bins contain less than 10 Z+ cells. Data are presented as the mean ± s.e.m.

The online version of this article includes the following source data and figure supplement(s) for figure 6:

**Source data 1.** *Figure 6B*: Frequencies of CXCR5, Bcl6 and IL-21 producers in respective subpopulations.
**Source data 2.** *Figure 6C*: Frequencies of IL-21 of CD44+ producers in respective subpopulations.
**Figure supplement 1.** PRI results can be confirmed with viSNE and conventional analysis.
**Figure supplement 1—source data 1.** *Figure 6—figure supplement 1A*: Frequencies of protein expressions sub-divided into regions.

(*Figure 7B*) which was reduced either by blocking IL-21 or CD40L signaling and was abrogated by blocking both pathways (*Figure 7B*). We could rule out B cell helping activity of high IFN-γ levels of these cells, as reported for IgG2a induction (*Snapper and Paul, 1987*), since the blockade of IFN-γ did not diminish total IgG production (*Figure 7B*).

To measure the relative helper functions of Tsh cells, we compared their ability to provide B cell help with that of other T cell subsets in vitro (*Figure 7C—figure supplement 2*). We sorted (CD4+ CD44+ PD-1[hi]) cells containing CXCR5+ Tfh helper cells with well-established helper functions (*Vinuesa et al., 2016*), Tsh (CD4+ CD44+ CXCR5- PD-1[low] CXCR3+), and as control population also the CD4+ CD44+ CXCR5- PD-1[low] CXCR3- subset, and co-cultured them with B cells in vitro for 5

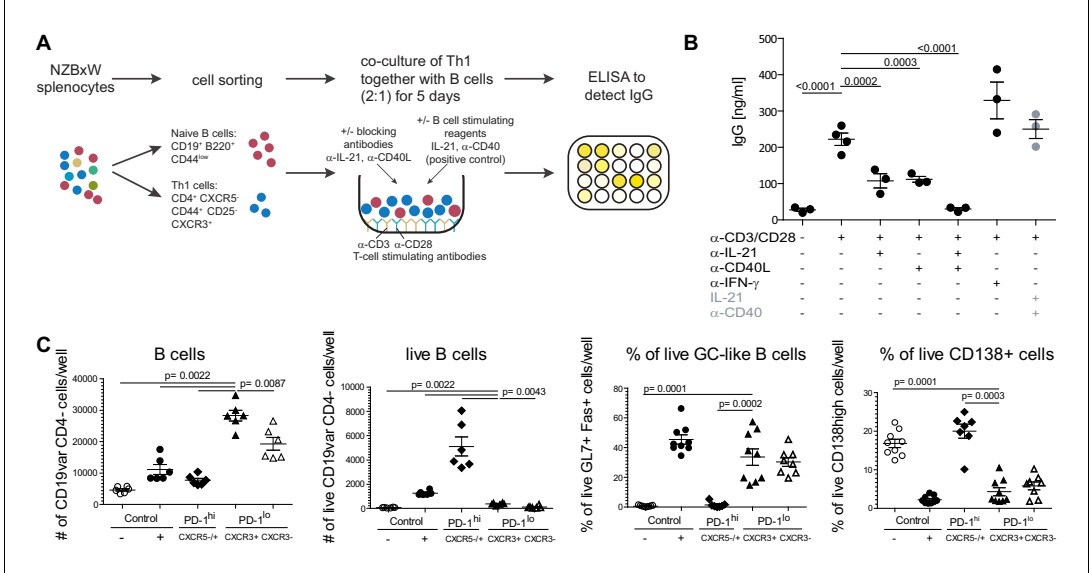

**Figure 7.** Super-functional T cells provide help for IgG production in co-cultures with B cells and expand B cells with activated GL7[+] Fas[+] Germinal Center-like phenotype. (**A**) Splenocytes were FACS-sorted for naïve B cells (CD19[+] B220[+] CD44[low]) and Th1 cells (CD4[+] CXCR5[-] CD44[+] CD25[-] CXCR3[+]). Both fractions were co-cultured for 5 days in presence or absence of antibodies against B cell activating cytokines. T cells were stimulated with agonistic antibodies against CD3 and CD28. As positive control, B cells were stimulated with IL-21 and an agonistic anti-CD40 antibody. IgG in the supernatant was detected by an ELISA. (**B**) ELISA results. One representative experiment of three is shown. Statistical analysis was performed by one-way ANOVA with Dunnett's multiple comparison test. (**C**) Total CD19[+] B220[+] B cells were co-cultured for 5 days either alone (negative control, -), or with IL-21 + anti-CD40 antibody (positive control, +), CXCR3[+] PD-1[lo] Tsh (PD-1[lo] CXCR3[+]), CXCR3[-] PD-1[lo] CD4 T cells (PD-1[lo] CXCR3[-]), or PD-1[hi] CXCR5[+/-] cells (PD-1[hi]) FACS-sorted from pooled splenocytes of 2-year-old C57Bl/6 mice. At day 5, viable B cells were identified and percentage of activated Germinal Center (GC)-like B cells was determined by co-expression of GL7 and Fas, antibody-producing plasma cells (PC) by expression of CD138. Data on B cell count and viability are representative of one out of two experiments involving six replicates per condition. Data on GL7[+] Fas[+] GC-like B cells and CD138[+] antibody-producing cells are pooled data from two independent experiments involving 1–7 replicates. Statistical significance of differences between CXCR3[+] PD-1[lo] Tsh co-cultures and other conditions were assessed by Mann Whitney test.

The online version of this article includes the following source data and figure supplement(s) for figure 7:

Source data 1. *Figure 7B*: Frequencies of IgG concentrations in co-cultures with different antibody settings.
Source data 2. *Figure 7C*: Cells per well and frequencies of GC-like and CD138[+] cells in co-cultures with different T cell subsets.
Figure supplement 1. Sorting strategy and purity control of naive B cells and Th1 cells.
Figure supplement 2. Functional comparison of CXCR3[+] PD-1[lo] Tsh, CXCR3[-] PD-1[lo] CD4[+] T cells and PD-1[hi] cells in B and T cell co-cultures.

days (*Figure 7—figure supplement 2A–B*). To obtain enough T cells, in particular PD-1[hi] cells that represent a minute population in NZBxW mice (*Figure 3C*), we resorted to old (15–24 months) C57BL/6 mice readily available and known to develop autoimmune features (*Hayashi et al., 1989*; *Nusser et al., 2014*). We had found that these mice also present with increased numbers of Tsh cells co-producing high levels of IFN-γ, IL-21 and CD40L (*Figure 7—figure supplement 2C*). At the end-point of the co-culture, we analyzed the B cells for their numbers, viability and phenotype by flow cytometry (*Figure 7—figure supplement 2D*). Strikingly, PD-1[low] CXCR3[+] Tsh cells promoted larger expansion of B cells than any other conditions, even outnumbering the positive control (IL-21 and anti-CD40) and PD-1[hi] cells by a factor of two and three, respectively (*Figure 7C—figure supplement 2D*). However, PD-1[hi] cells were clearly superior in preserving more viable B cells than the other conditions (*Figure 7C—figure supplement 2D*). The phenotype of B cells cultured with Tsh cells was in stark contrast compared to that of B cells cultured with PD-1[hi] cells. While the proportion of CD138[+] antibody-producing B cells was higher in the co-cultures with PD-1[hi] cells (20% against 2–6% in other cultures), Tsh cells induced 33% of the B cells to express the activation markers GL7 and Fas, typically used to define germinal center (GC) B cells, against 1.5% of B cells with a GC-like phenotype in PD-1[hi] co-cultures (*Figure 7C—figure supplement 2D*). The proportion of GC-like B cells recovered with Tsh cells was similar to that found with PD-1[low] CXCR3[-] cells and the positive control (*Figure 7C—figure supplement 2D*). Nevertheless, given the superiority of Tsh cells to promote B

cell expansion, this experiment indicates that Tsh cells have highly potent helper capacity and suggests complementary functions to that of PD-1$^{hi}$ and Tfh cells during B cell differentiation.

All this suggests that the IL-21 producing super-functional Th cells (IL-21$^+$ IFN-$\gamma^{hi}$ PD-1$^{low}$ Tmem cells) represent a novel Tfh-like subpopulation that can provide B-cell help in lupus nephritis mice in an IL-21- and CD40L-dependent manner.

## Discussion

For many autoimmune diseases, the underlying cellular mechanisms are still incompletely understood. To develop novel targeted therapies, it is important to identify potentially pathogenic immune cell subsets. To this end, appropriate methods for the analysis and visualization of high-dimensional cytometric data are crucial to document global changes in the immune cell pattern during the course of the disease. Our study has generated three important findings: it showed the added value of our PRI-visualization, it identified a super-functional Tfh-like cell subpopulation and it indicated its potential functional effect on SLE.

First, PRI enabled the identification and characterization of the super-functional IL-21$^+$ IFN-$\gamma^{hi}$ cell subset by its main functionalities: (i) simple visualization of the combinatorial properties of three markers and heat map representation of different statistical attributes of the third marker by binning conventional two-parametric dot plots (e.g. *Figure 1E*, *Figure 6E*); (ii) semi-continuous, combinatorial display that allows straightforward perception of correlating patterns (e.g. *Figure 3D–E*); (iii) reproducible pseudo-multi-parametric display of different markers so that subpopulations can be intuitively observed and compared, even between samples of different staining panels, which is not applicable with cluster algorithms (e.g. *Figure 4A and B*; *Qiu et al., 2011*; *Shen et al., 2018*) and (iv) exploitation of auxiliary information in percentages is enabled by the combinatorial approach, which would otherwise only be accessible with further gating steps. It should also be noted, however, that with PRI many cells and a broad bin range together with the right marker combination are necessary to obtain meaningful information from the plots.

PRI paves the way for reproducible and automatable cytometric analyses, which are useful not only for basic research but also for clinical studies to identify disease mechanisms and progressions, to discover biomarkers, to analyze therapy responder/non-responder, and to monitor stable quality and effectiveness of manipulated cells. In addition, PRI provides a statistically robust pattern map for visual inspections and serves as input for prospective pattern recognition in machine learning approaches with feature engineering steps.

Second, the super-functional IL-21$^+$IFN-$\gamma^{hi}$ cell subset identified here is a novel Tfh-like subpopulation, which differs from the subpopulations described so far by a low expression of PD-1. In concordance with Tfh, Tph (*Rao et al., 2017*) and other Tfh-like cell subsets (*Bubier et al., 2009*; *Hutloff, 2018*; *Vu Van et al., 2016*; *Weinstein et al., 2016*) splenic super-functional T cells are IL-21$^+$, CD40L$^{hi}$, and ICOS$^+$ (*Figure 4—figure supplement 1*). Their affiliation to Tfh or extrafollicular Tfh cells could be ruled out, as the majority of super-functional T cells do neither express Bcl6 or CXCR5, nor CXCR4 or PSGL-1$^{low}$ (*Figure 4*; *Odegard et al., 2008*). They rather seem to exhibit polyfunctional characteristics such as a strong co-expression of Th1 and Tfh-like effector proteins with high activation potential.

Poly-functional T cells were first described as being protective in viral infections (*Kannanganat et al., 2007*; *Seder et al., 2008*). However, polyfunctional T cell subpopulations were identified as pathogenic in autoimmune diseases (*Basdeo et al., 2015*). Most of the previous studies on poly-functional T cells focused on IFN-$\gamma^{hi}$, TNF-$\alpha^{hi}$, IL-2$^{hi}$ cells in the context of infections and immunizations (*Mateus et al., 2019*; *Sauzullo et al., 2014*; *Westerhof et al., 2019*). Compared to the triple positive poly-functional T cells, super-functional T cells here were even superior regarding their additional expression of IL-21, ICOS, and CD40L$^{hi}$ and their demonstrated B cell helper activity (*Figure 7*). In vitro, PD-1$^{low}$ CXCR3$^+$ CXCR5$^-$ Tsh cells and PD-1$^{hi}$ cells, containing the CXCR5$^+$ Tfh cells, displayed dichotomic functions regarding the induction of GC-like and plasma cell phenotypes, respectively. In keeping with the chemokine profile of these two CD4 T cell subsets, this result suggests that in vivo PD-1$^{low}$ CXCR3$^+$ CXCR5$^-$ Tsh cells could provide help in the inter-follicular space juxtaposed to the GC and enriched in the CXCR3 ligand CXCL9 (*Matloubian and Cyster, 2012*). There, B cells become blasts, proliferate and differentiate into GC B cells that migrate to CXCR5 ligand-rich follicles, where further support by the Tfh promotes affinity maturation and plasma cells

differentiation (*Haynes et al., 2007*; *Vinuesa et al., 2016*). Alternatively, Tsh could participate to expand activated B cells in an extrafollicular GC-like reaction, as recently described during Salmonella infections (*Di Niro et al., 2015*).

Considering the observed extrafollicular co-localization of T cells and B cells in the spleen of NZBxW mice (data not presented) and the peripheral localization of super-functional T cells (*Figure 5D*), we speculate that they can even provide B cell help directly in inflamed tissues, thereby exacerbating tissue destruction. Due to their low expression of inhibitory receptors (especially PD-1, CTLA4), their activation is potentially facilitated compared to Tfh cells, as these express significantly higher amounts of PD-1, CTLA4, TIGIT and BTLA (*Figure 4*; *Butte et al., 2007*; *Walunas et al., 1994*).

Third, the super-functional IL-21$^+$IFN-$\gamma^{hi}$ cell subset could promote SLE progression through B cell help dependent and independent effects. Similar to Tfh-like cells, which were found in SLE and other autoimmune diseases (*Horiuchi and Ueno, 2018*; *Hutloff, 2018*; *Rao, 2018*; *Vu Van et al., 2016*), the super-functional subset delivers effective B-cell help for IgG production in an IL-21- and CD40L-dependent manner. Tissue-resident Tfh-like cells are proposed to be 'the most pathogenic T cell subset, as they select autoreactive B cells in the uncontrolled environment of lymphocytic tissue infiltrates and drive the local differentiation of plasmablasts producing pathogenic antibodies directly in the affected tissues' (*Hutloff, 2018*).

Due to their presence also in non-lymphoid organs (*Figure 5*) and their IL-21 and IFN-$\gamma$ co-expression, the super-functional cell subset might be superior in activation-induced processes that drive autoimmunity by expanding pathogenic T cell subsets in non-lymphoid tissue and promoting inflammation. This hypothesis is supported by several studies focusing either on IL-21 or IFN-$\gamma$ effects.

IL-21 uniquely contributes to SLE and other autoimmune diseases, because pharmacological and genetic abrogation of IL-21 signaling in mice stopped inflammation in non-lymphoid organs and protected them from autoimmune diseases, respectively (*Choi et al., 2017*; *Liu and King, 2013*). In line with the reported IL-21–mediated proliferation (*Zeng et al., 2007*) and its autocrine and paracrine action (*Nurieva et al., 2007*) is our observation that the IL-21$^+$ IFN-$\gamma$+ double-producing population had the highest proliferative capacity in vivo compared to single-producing (IL-21$^+$ IFN-$\gamma^-$, IL-21$^-$ IFN-$\gamma$+) and double-negative (IL-21$^-$ IFN-$\gamma^-$) populations measured by Ki67 expression (data not shown).

IFN-$\gamma$ is also considered to be a key molecule in the pathogenesis of SLE (*Peng et al., 2002*; *Pollard et al., 2013*; *Tsokos et al., 2016*). Consequently, excessive IFN-$\gamma$ production was required to sustain lupus-associated Tfh cell accumulation in mice (*Lee et al., 2012*). Contrary to this and other reports on lupus-prone mice (*Enghard et al., 2006*; *Humrich et al., 2010*), we observed no increase but a slight decrease in IFN-$\gamma$ frequencies with disease progression (*Figure 2*), possibly due to different subpopulations of Tmem and Th cells being studied.

In addition to IL-21 and IFN-$\gamma^{hi}$ co-expression, the super-functional cell subset is characterized by high IL-2 and TNF-$\alpha$ but no IL-10 expression. IL-2 is discussed to be rather protective than pathogenic in SLE because the acquired IL-2 deficiency is a crucial general event in the pathogenesis of lupus and mainly leads to a disorder of the homeostasis of regulatory T cells. Correspondingly, a low dose IL-2 therapy selectively corrected Treg cell defects and greatly expanded the Treg cell subpopulation (*Humrich et al., 2015*; *von Spee-Mayer et al., 2016*). In contrast, anti-TNF-$\alpha$ therapy with infliximab has produced inconsistent results so far and is not routinely used to treat SLE (*Aringer and Smolen, 2012*). IL-10 has pleiotropic effects, which can be anti-inflammatory and pro-inflammatory. In SLE patients, however, the higher IL-10 level associated with the disease is considered pathogenic and its blockage improves the disease (*Llorente et al., 2000*). Recently, an IL-10/IFN-$\gamma$ co-producing pathogenic Th cell sub-population (CXCR5$^-$ CXCR3$^+$ PD-1$^{hi}$) was identified in SLE patients (*Caielli et al., 2019*). These cells differ from our super-functional cells primarily because they do not co-express IL-21, rather they provide B cell help independently of IL-21, and are PD-1 high.

Transient co-expression of IL-21 and IFN-$\gamma$ has already been described for subsets of early Th1-Tfh and germinal centre Tfh cells (*Fang et al., 2018*; *Nakayamada et al., 2011*; *Weinstein et al., 2016*). However, it is still unclear to what extent double producing cells functionally differ from IFN-$\gamma$ and IL-21 single producers (*Song and Craft, 2019*). It is worth mentioning in this context that the majority of IL-21$^+$ Tmem cells in this study were IFN-$\gamma$+ (*Figure 3F*; young: 56 ± 3% versus old

diseased: 71 ± 4%) and non-Tfh cells (*Figure 6*). They were found in both lymphoid and peripheral organs of NZBxW mice (*Figure 5*).

Interestingly, IL-21 was reported to inhibit TCR-induced Th1 differentiation (*Kastirr et al., 2014*; *Suto et al., 2006*; *Wurster et al., 2002*) and, vice versa, forced expression of the Th1 master transcription factor T-bet inhibited IL-21 expression (*Kastirr et al., 2014*). Therefore, the special co-expression of IL-21 and IFN-γ could be triggered by chronic inflammatory conditions, leading to a dysregulated cytokine landscape (*Humrich et al., 2010*), specific expansion of double positive cells (*Kastirr et al., 2014*), and plasticity of Th cell phenotype (*Carpio et al., 2015*).

Several reports underline a high diversity and plasticity of Tfh and Tfh-like cells (*Lüthje et al., 2012*; *Song and Craft, 2019*; *Wong et al., 2015*). Song and Craft even proposed in between the two archetypes are differentiated cells that have varied degrees of Th and Tfh cell characteristics which may not exist in distinct subsets" (*Song and Craft, 2019*). They proposed to search for alternatives to further dividing Th, Tfh and Tfh-like cell subsets. Our PRI approach is an attempt in this direction to analyze and visualize a continuum between cell subsets without further dividing subpopulations by gating (*Figure 4*, *Figure 3—figure supplement 1D*).

In this study, we defined a novel subset of Tfh-like cells – super-functional IL-21$^+$ IFN-γ$^{hi}$ PD-1$^{low}$ Tmem cells – that exhibit superior production of cytokines in both quantity and number and provide B cell help. This suggests that efforts to decrease their frequency or manipulate their main properties such as IL-21/IFN-γ co-expression may represent an important therapeutic strategy.

## Materials and methods

### Mice

NZBxNZW (NZBxW) mice were bred under specific-pathogen-free conditions in the animal facility of the Federal Institute for Risk Assessment (Berlin, Germany). The female filial one generation was used for experiments. Animal experiments were approved by the local authority LAGeSo (Landesamt für Gesundheit und Soziales) Berlin under animal experiment licenses T0187-01 and G0070/13.

Severity of disease was monitored weekly by scoring the mice according to levels of proteinuria (Uristix, Siemens) and weight.

C57Bl6/J mice were purchased from Janvier (France) and kept for 15 to 24 months in the SPF facility of the DRFZ.

### Cell preparations for flow cytometry

We isolated splenocytes by crushing the spleen through a 200 µm metal mesh in order to create single-cell suspensions. Erythrocytes were lysed by hypotonic shock and splenocytes were filtered through a 30 µm cell strainer.

Kidney, liver and lung were perfused with PBS. From the liver, single-cell suspensions were created by crushing through a 200 µm metal mesh. To isolate hepatic lymphoid cells, a density gradient centrifugation in the presence of 40% Percoll was performed followed by erythrocyte lysis.

Lungs and kidneys were perfused with PBS and placed in ice-cold PBS. Organs were cut into small pieces and digested with 0.25 mg/ml collagenase (Sigma) and 0.25 mg/ml collagenase D (Roche) and 10 U/ml DNAse I (Sigma) in RPMI 1640 medium + 0.5% BSA. The tissue was gently crushed through a 200 µm metal mesh followed by a 70 µm cell strainer. Cells were washed and resuspended in RPMI1640 supplemented with 10% FCS, b-mercaptoethanol, penicillin (100 U/ml) and streptomycin (100 U/ml).

### Flow cytometry

The analysis was conducted according to the guidelines for the use of flow cytometry and cell sorting in immunological studies (*Cossarizza et al., 2017*).

For intracellular cytokine staining, cells were stimulated with 10 ng/ml phorbol 12-myristate 13-acetate (PMA) and 1 µg/ml ionomycin (Sigma) for 1 hr followed by additional 3 hr in the presence of 5 µg/ml Brefeldin A (Sigma). Cell surfaces were stained with antibodies in the presence of 100 µg/ml 2.4G2 (anti-FcγRII/III; purified from hybridoma supernatants) to reduce unspecific antibody binding. Cells were fixed with 2% paraformaldehyde followed by an intracellular staining in 0.5% saponin. For intracellular staining of transcription factors, cells were fixed with FoxP3 fixation buffer (eBioscience)

and stained in FoxP3 permeabilization buffer (eBioscience). Live cells were discriminated from dead cells with a fixable LIVE/DEAD stain (Life Technologies). The expression of phenotypic markers was determined with a BD LSR Fortessa (BD Biosciences) and analyzed with FlowJo (Treestar).

## Detection of serum antibodies by enzyme-linked immunosorbent assay (ELISA)

Serum was collected from mice of different age. Anti-dsDNA-IgG antibodies were measured by ELISA as described previously using biotin-labeled goat anti-mouse IgG (γ chain specific) antibodies for detection (Southern Biotech) (*Cheng et al., 2013*).

## Co-culture experiments

Splenocytes were stained with anti-CD19-FITC, anti-CD25-PerCP-Cy5.5, anti-CXCR5-PE-Cy7, anti-CXCR3-APC, anti-B220-APC-Cy7, anti-CD4-PacB, anti-CD44-BV785, DAPI and sorted for DAPI⁻ naive B cells (CD19$^+$ B220$^+$ CD44$^{low}$) and Th1 cells (CD4$^+$ CXCR5$^-$ CD44$^+$ CD25$^-$ CXCR3$^+$) using FACS Aria II (BD Biosciences) with a purity ≥95%. Th1 cells and naive B cells (2:1) were sterily co-cultured for five days in 96-well round-bottom plates in the presence or absence of blocking antibodies: IL-21R-Fc chimera (R and D systems), anti-CD40L (clone MR1, DRFZ) and anti-IFN-γ (clone AN18.17.24, DRFZ). T cells were stimulated polyclonally with anti-CD3 (clone KT3) and anti-CD28 antibodies (clone 37.51, both DRFZ). As positive control, B cells were stimulated with IL-21 (Pepro-tech) and anti-CD40 (clone FGK-45, DRFZ). After 5 days, the IgG concentration in the supernatant was determined by the Mouse IgG ELISA Kit (LSBio).

T cell populations for T helper functions' comparison were FACS-sorted from splenocytes stained with anti-PD-1 conjugated to PercP-Cy5.5 or APC-Cy7, anti-CXCR5-BV650, anti-CD44 conjugated with PacO or BV786, anti-CXCR3 conjugated with APC or BV421, anti-CD4-PE, anti-CD8-Al488, DAPI or PI. B cells were sorted from splenocytes after enrichment with B cell purification kit using magnetic beads (Miltenyi, Germany), and stained with anti-B220-FITC, anti-CD19-Al647 and PI. PI⁻ CD19$^+$ B220$^+$ B cells, live CD4+ PD-1lo CXCR5- CXCR3+ or CXCR3- and PD-1hi CXCR5+/- were sorted using FACS Aria II (BD Biosciences) with a purity ≥90%. T and B cells were cultured as above. On day five, cells were harvested, washed in PBS, and stained with L/D aqua fixable dye (Thermo-FisherScientific) and with anti-B220-FITC, anti-CD4-PE, anti-GL7-PerCP-Cy5.5, anti-Fas-PE-Cy7, anti-CD138-BV421. For each well, the totality of the cells were acquired on BD LSR Fortessa (BD Biosciences) and analyzed with FlowJo (Treestar).

## Flow cytometry analysis

Files were pre-processed in FlowJo (compensation, elimination of doublets and dead cells, gating). For tSNE, CD44$^+$ T cells were down-sampled to 10.000 cells and analyzed with the plugin 'Tsne' (1000 iterations, perplexity 20, theta 0.5, eta 200) in FlowJo. For PRI, T cell populations were exported as fcs-files. Fluorescent intensities were transformed with inverse hyperbolic sine (arcsinh), which is a common transformation method for flow cytometry data (*Finak et al., 2010*). The x and y axis of parameters X *vs.* Y was categorized into bins of size $0.2 \times 0.2$. Over all cells in each bin, different statistical methods are calculated and plotted in a color-coded manner as a heat map (low values are represented by shades of blue, median values by yellow and high values by red). The frequency (%) of parameter Z$^+$ cells per bin, mean fluorescence intensity of parameter Z$^+$ cells (MFI +), MFI over all cells per bin or the relative standard error of the mean of parameter Z (RSEM) were used. Auxiliary information are displayed as black, red, green and blue percentage numbers in each quadrant. Black percentage numbers indicate the frequency of cells relative to total cells. The frequency of Z$^+$ cells is given as relative to cells inside the respective quadrant (red), relative to total cells (green) and relative to total Z$^+$ cells (blue). Frequencies of double producing cells (two parameters) per bin are color-coded in green scales. All parameters were included in the analysis and bins with less than 10 cells are not displayed. This minimum count of cells was used to balance the impact of identifying comparatively rare subpopulations while retaining the statistical robustness. We operated on R with package 'flowCore' to read the fcs-files. In order to generate 3D-surface plots with package plotly v4.7.1, frequency bin tables were exported from PRI's R source code.

## Quantification and statistical analysis

Statistical analyses were conducted using GraphPad Prism7 Software. For bar and scatter plots, data are shown as mean ± s.e.m. The statistical tests included unpaired, two-tailed Student's t-test and Mann-Whitney test. For comparisons of three or more groups, data were subjected to one-way or two-way ANOVA, followed by Sidak's or Dunnett's multiple comparison test (as indicated in each figure legend). $p < 0.05$ was considered significant.

## Acknowledgements

We thank all members of the lab, and Tobias Alexander, Gabriele Riemekasten, and Jens Humrich for helpful comments and discussions. We thank Falk Hiepe and Manuela Frese-Schaper for providing NZBxW mice. We are grateful to Jenny Kirsch and Toralf Kaiser for operating the flow cytometry core facility (FCCF) and also to the staff of the animal facility and the lab managers.

## Additional information

### Funding

| Funder | Grant reference number | Author |
| --- | --- | --- |
| Bundesministerium für Bildung und Forschung | 0316164A | Ria Baumgrass |

The funders had no role in study design, data collection and interpretation, or the decision to submit the work for publication.

### Author contributions

Stefanie Gryzik, Conceptualization, Data curation, Formal analysis, Validation, Investigation, Visualization, Methodology, Writing - original draft; Yen Hoang, Conceptualization, Data curation, Software, Formal analysis, Validation, Investigation, Visualization, Methodology, Writing - original draft; Timo Lischke, Conceptualization, Investigation, Methodology; Elodie Mohr, Formal analysis, Investigation, Methodology, Writing - review and editing; Melanie Venzke, Resources; Isabelle Kadner, Software; Josephine Poetzsch, Investigation; Detlef Groth, Methodology; Andreas Radbruch, Funding acquisition; Andreas Hutloff, Supervision, Methodology, Writing - review and editing; Ria Baumgrass, Conceptualization, Resources, Supervision, Funding acquisition, Investigation, Writing - original draft, Project administration

### Author ORCIDs

Yen Hoang  https://orcid.org/0000-0001-8956-1709
Timo Lischke  http://orcid.org/0000-0003-0413-4252
Elodie Mohr  https://orcid.org/0000-0003-3406-7302
Josephine Poetzsch  https://orcid.org/0000-0003-4238-3866
Andreas Hutloff  https://orcid.org/0000-0002-0572-8151
Ria Baumgrass  https://orcid.org/0000-0002-3289-1608

### Ethics

Animal experimentation: Animal experiments were approved by the local ethics committee LaGeSo (Landesamt für Gesundheit und Soziales) Berlin under animal experiment licenses T0187-01 and G0070/13.

### Decision letter and Author response

Decision letter https://doi.org/10.7554/eLife.53226.sa1
Author response https://doi.org/10.7554/eLife.53226.sa2

## Additional files

### Supplementary files

- Source code 1. A notebook to show step by step how to create the bin plots (in HTML).
- Source code 2. A notebook how to create bin functions (in R).
- Supplementary file 1. Key resources table.
- Transparent reporting form

### Data availability

Flow cytometry data have been deposited in FlowRepository under the accession code FR-FCM-Z2C8. All data generated or analysed during this study are included in the manuscript and supporting files. Source data files have been provided for Figure 1D (Figure 1-source data 1); Figure 2A,B,D, E (Figure 2-source data 1-4) and Figure 2-figure supplement 1B (Figure 2-figure supplement 1-source data 1); Figure 2-figure supplement 2B,E (Figure 2-figure supplement 2-source data 1-2); Figure 2-figure supplement 3D (Figure 2-figure supplement 3-source data 1); Figure 3C (Figure 3-source data 1); Figure 3-figure supplement 1A-C (Figure 3-figure supplement 1-source data 1-2); Figure 5A-D (Figure 5-source data 1-4); Figure 6B,C (Figure 6-source data 1-2) and Figure 6-figure supplement 1A (Figure 6-figure supplement 1-source data 1); and Figure 7B,C (Figure 7-source data 1-2).

The following dataset was generated:

| Author(s) | Year | Dataset title | Dataset URL | Database and Identifier |
|---|---|---|---|---|
| Gryzik, Hoang | 2019 | PRI Tfh-like | http://flowrepository.org/id/FR-FCM-Z2C8 | Flow Repository, FR-FCM-Z2C8 |

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
