## [Decision Letter]

**Acceptance summary:**

Some T helper cell populations have been shown to expand in autoimmune diseases such as systemic lupus erythematosus and might thus contribute to the unfolding of the disease condition. To zoom on those T cells that have phenotype very similar to other T cell subpopulations, the authors combined a bin-based "pattern recognition of immune cells" strategy with comprehensive cytometric measurements of T helper cells from found in lupus-prone mice. On that basis, they defined a novel subset of super functional T helper cells that exhibit superior B cell help and may constitute therapeutic targets.

**Decision letter after peer review:**

Thank you for submitting your article "Identification of a super-functional Tfh-like subpopulation in murine lupus by pattern perception" for consideration by *eLife*. Your article has been reviewed by three peer reviewers, and the evaluation has been overseen by a Reviewing Editor and Satyajit Rath as the Senior Editor. The following individual involved in review of your submission has agreed to reveal their identity: Joe Craft (Reviewer #1).

The reviewers have discussed the reviews with one another and the Reviewing Editor has drafted this decision to help you prepare a revised submission.

Summary:

The authors used CytoBinning, a recently described flow cytometry analysis approach, to explore the phenotype and cytokine expression profile of splenic memory CD4^+^ T cells from NZBxW mice at different stages of disease. The analysis approach combines automation of a traditional workflow and machine learning, linking high dimensional data back to two biomarkers which can be represented as 2D scatter plots and enables the identification of subtle shifts in immune phenotypes. Using this approach, the authors identify a subpopulation of IL21^+^ IFNγ^+^ PD1^low^ CD40L^high^ CXCR5^-^ Bcl6^-^ CD4^+^ T cells expanded in older, diseased mice that expresses high levels of TNFα and IL2. Considering that in co-culture experiments they observe that it provides B cell help though IL21 and CD40L, the authors define this population as a "super-functional Tfh-like cells". Overall, this interesting manuscript surmises that these newly identified cells are critical for the disease phenotype in lupus, including promotion of autoantibody production.

As a major aim of the manuscript is the presentation of a new method to analyse flow cytometry data, further information should be made available to allow a direct comparison of this new method with previous methods. Moreover, the "super-functional Tfh-like cells" should be more precisely compared to the T peripheral helper (Tph) cells, first described in rheumatoid arthritis by Rao and colleagues (Rao et al., 2017), and this year in SLE (Bocharnikov et al., 2019), celiac disease (Christophersen et al., 2019) and T1DM (Ekman et al., 2019).

Essential revisions:

1) Methodological issues.

1. 1) While the insights revealed using the combinatorial approach with data binning are of interest, sifting through the data is more challenging than it need be. Certain panels of figures are mislabeled (2H does not exist, 3C or D is incorrectly noted, and so forth), and the authors are asked to proof the text and figures. As well, Figure 2E and Figure 2—figure supplement 1B are nigh on impossible to parse, given overlap in colors (not even accounting for red-green discernment). In a like vein, certain figures are challenging to interpret, given the light shades of gray that appear to overlap. The concerns stand out all the more, given the nice visuals of the binning data.

1.2) Each PRI plot superimposes data from several samples (biological replicates). While the authors show that data are reproducible for individual mice, it is likely that for some markers or some biological experiments, great inter-individual variation can occur. The provided statistics fail to represent this issue. A better clarification on how PRI plots can be used to show the range of variability among a population should be provided.

1.3) The audience of this manuscript will be far more familiar with traditional methods of FACS analysis than PRI. As a consequence, in order to establish the robustness of PRI-based quantification and familiarity with the approach, Figure 3A should include also samples from young mice. It seems from the analysis of the flow cytometry plots of old mice (Figure 3A) that all IL-21^+^ cells co-express IL-2, TNF, and IFN while being CXCR5^-^, PD-1int, and IL-10^-^. However, PRI assessment of the same samples (Figure 3F) provides a visual representation suggesting that the vast majority of IFNγ^+^ cells are also producing IL-21, while in the dot plots (Figure 3A) only ~1/7 of IFNγ^+^ cells produce IL-21. Furthermore, looking at the PRI of young mice it is unclear whether the IL-21 producing cells maintain the same phenotype as the cells from old mice, like the authors claim (subsection “Bin plots revealed an IL-21^+^ super-functional T cell subpopulation”). For all of the above Figure 3A should allow independent assessment of young/old phenotype in addition to PRI plots of Figure 3F.

1.4) Additionally, the most biologically relevant statistical assessment is in respect to cell populations (defined to the best of our ability) rather than individual markers. If the authors claim the importance of a novel cell subset defined as IL-21^+^, IFNγ^+^… the relevant statistical analysis of Figure 3E should be based on the frequency of that cell subset in young vs. old mice rather than (or in addition to) the presented analysis based on individual markers.

1.5) The phenotype of IL-21-producing cells displayed in Figure 4 is not easy to interpret given the poor identification of the IL21^+^ cells, as they are contained in bins together with other cells. Lack of statistics is also a concern. As a key objective of the manuscript is to show how PRI compares with FACS analysis, the figure would improve with traditional dot plots of gated IL-21^+^ cells (or even how IL21^+^IFNγ^++^ compared with IL-21^-^IFN^++^ for the different markers – to a large extent these two cell populations are in the same bin region). Statistics could then be provided and compared with PRI data.

1.6) It is also puzzling why the authors do not compare the phenotype of IL-21^+^ cells and IL-21^-^IFN^+^ and IL21^-^IFN^-^ instead of a complex matrix of comparisons based on IFNγ and PD-1 expression (Figures 5, 6 and Figure 6—figure supplement 1). A major goal of the manuscript is to define a IL-21^+^IFNγ^+^ cell population that can be easily gated. Data should be provided for the population of interest in addition to arbitrary quadrants.

1.7) Figure 2C. How was the determination of PD-1^-^, PD-1^low^ and PD-1 positivity determined? The gating does not follow the cell density plots.

1.8) The authors note the data are reproducible. but how was reproducibility shown statistically?

2) Biological issues

2.1) The authors focused only on the "super-functional Tfh-like cells", even though in the sicker mice there is not only and expansion of IL21^+^ IFNγ^+^ PD1^low^ cells but also of IL21^+^ IFNγ^+^ PD1^high^ cells (Figure 3E) as well IL10^+^ IFNγ^+^ PD1^high^ cells. To prove that this IL21^+^ INFγ^+^ PD1^low^ CD40L^high^ CXCR5^-^ Bcl6^-^ T cell population helps B cells though IL21 and CD40L, they sort non Tfh-Th1 like cells (CD4^+^, CXCR5^-^, CD44^hi^, CXCR3^+^ cells) and check their ability to support B cell activation and Ig production. Two main concerns arise from these set of experiments. First, the sorting strategy should include PD1 in order to enrich the non Tfh-Th1 like mem T cells (CD4^+^, CXCR5^-^, CD44^hi^, CXCR3^+^) within the PD1^low^ population (CD4^+^, PD1^low^, CXCR5^-^, CD44^hi^, CXCR3^+^ cells). Second, the B cell helper assay should be repeated and compared with other cell populations, including conventional Tfh1 cells (CD4^+^, PD1^hi^, CXCR5^+^, CD44^hi^, CXCR3^+^ cells) and include naïve versus total B cells. In addition, the authors should perform a more thorough characterization of the main three expanded cell subpopulations by RNAseq to better define the uniqueness of the novel "super- functional" cells.

2.2) The authors claim IFNγ^+^IL-21-producing cells represent a novel population distinct from previous defined populations with "super-functional" properties.

2.2.1) One cell population that has been shown important in human SLE and bears some similarities was not discussed: T peripheral helper (Tph) cells, first described in rheumatoid arthritis by Rao and colleagues (Rao et al., 2017), and this year in SLE (Bocharnikov et al., 2019), in celiac disease (Christophersen et al., 2019) and T1DM (Ekman et al., 2019). These cells are CXCR5^-^ Bcl6^-^ and characterized by IL-21 production, a critical difference to the population discussed in this manuscript is that Tph are PD-1^hi^. However, if the two populations are different, they should both be present. Do the authors claim the only T cell population producing IL-21 is the newly discovered? Is there an explanation for poor identification of other IL-21-producing T cells among CXCR5^-^ cells, namely Tph?

2.2.2) Considering that the super-functional T cell population identified by the authors is similar to Tph cells, it would be of interest to know if the super-functional T cells are Blimp1^+^, which is likely the case, recognizing the potential limitation of flow staining for this transcriptional repressor in mouse.

2.2.3) While the T-B cell helper assay adds a functional component to the paper, how physiological this experiment really is? Do the authors think that their T cells drive "naïve" B cells to produce Ig in vivo? Do super-functional T cells come into contact with naïve B cells in secondary lymphoid organs or elsewhere? Wouldn't naïve B cells need an IgM as well as TLR and/or cytokine signals first, to move into proximity with activated T cells?

2.2.4) The functional assay cannot be easily interpreted with the available information. Since sorting was done with an indirect strategy (based on CXCR3^+^CXCR5^-^ T cells) it will be important to show what is the frequency of the IL-21^+^IFNγ^+^PD-1int cells among the sorted population. A positive control with bona fide CXCR5^+^PD1^hi^ Tfh cells should be provided to compare the potency of Tfh and the claimed "super-functional" population. Titrations may be provided to assess relative potency, if needed. It is hard to claim super-functional properties without a functional comparison with the current standard of B cell help (i.e. Tfh cells). FACS plots with B cells stained with anti-GL7 and anti-IgG1 could also make the results more compelling, given the low levels of IgG antibodies produced.

2.2.5) Frequently, BCR agonists (anti-IgM for instance) are provided in addition to T cell stimulation for in vitro assays (see for instance Sage, Nature Immunol 2016 17:1436), this may improve the assay read-out.

2.2.6) More clarity about disease definition would be helpful. "Chronic inflammation, high proteinuria", is not so specific, not is age of mice. Likewise, were autoantibodies and renal histology assessed?

---

## [Author Response]

Summary:The authors used CytoBinning, a recently described flow cytometry analysis approach, to explore the phenotype and cytokine expression profile of splenic memory CD4^+^ T cells from NZBxW mice at different stages of disease. […] Overall, this interesting manuscript surmises that these newly identified cells are critical for the disease phenotype in lupus, including promotion of autoantibody production.

Binning is a way of discretizing data and can be done by adaptive binning (e.g. quantiles) or fixed-width binning (e.g. rounding). The mentioned CytoBinning approach, first described by Roederer et al., 2001 (Cytometry 2001, 45:37), use the percentile-based binning. In contrast, our approach uses fixed-width bins with a varying number of cells. This visualisation was chosen, because the cells tend to accumulate near the center of the data range but flatten out fast near the boundary of the range. A percentile-based binning would emphasize on the cell distribution. In contrast, we focus on the expression distribution and pattern mainly at the range boundaries. We do not only use the regular cell count, but we particularly make use of the mean fluorescence intensities and the frequency of positive cells of an additional third marker or several markers to provide valuable additional information. This is the combinatorics of 3 or more markers. Furthermore, a fixed-width binning allows for a good intuitive comparison of different samples.

As a major aim of the manuscript is the presentation of a new method to analyse flow cytometry data, further information should be made available to allow a direct comparison of this new method with previous methods.

We accepted the advice and integrated additional conventional analyses into several figures. In Figure 3A and B we included all respective contour plots for young mice. We confirmed the bin plot data of new Figure 3F and G by conventional gating analysis shown in Figure 3A and B, and the results of statistical analysis in Figure 3C. For comparison we show the statistical analysis from the bin plots in Figure 3—figure supplement 1B.

Moreover, the "super-functional Tfh-like cells" should be more precisely compared to the T peripheral helper (Tph) cells, first described in rheumatoid arthritis by Rao and colleagues (Rao et al., 2017), and this year in SLE (Bocharnikov et al., 2019), celiac disease (Christophersen et al., 2019) and T1DM (Ekman et al., 2019).

Thank you for the recommendation. Tfh and Tph cells are both characterized by very high PD-1 expression (which distinguishes them from the PD-1^low^ super-functional Tfh-like (Tsh) cells), high production of IL-21, and either high or absent expression of CXCR5. We included new data and figures for these three markers (Figure 3C lower row, Figure 3G, Figure 3—figure supplement 1B and Figure 5D). In addition, we inserted additional references into the main text of the manuscript.

Essential revisions:1) Methodological issues.1. 1) While the insights revealed using the combinatorial approach with data binning are of interest, sifting through the data is more challenging than it need be. Certain panels of figures are mislabeled (2H does not exist, 3C or D is incorrectly noted, and so forth), and the authors are asked to proof the text and figures.

Sincerely sorry for the mistake. We corrected it.

As well, Figure 2E and Figure 2—figure supplement 1B are nigh on impossible to parse, given overlap in colors (not even accounting for red-green discernment). In a like vein, certain figures are challenging to interpret, given the light shades of gray that appear to overlap. The concerns stand out all the more, given the nice visuals of the binning data.

We totally see the point. We improved the colours and the graphics for both pie charts. For simplicity, we moved the pie chart (former Figure 2E) to Figure 2—figure supplement 2E).

1.2) Each PRI plot superimposes data from several samples (biological replicates). While the authors show that data are reproducible for individual mice, it is likely that for some markers or some biological experiments, great inter-individual variation can occur. The provided statistics fail to represent this issue. A better clarification on how PRI plots can be used to show the range of variability among a population should be provided.

In the original version of the manuscript, each bin plot visualised the cytometric data of a single mouse. To minimize inter-individual deviations, data from different mice were combined after measurement and were visualised in one bin plot. Therefore, for the new Figure 3 pictures we used concatenated cells from 3 mice each (young and old diseased). To exemplary demonstrate the robustness of the method, we compared the pattern of concatenated cells from the new experiments with patterns of 4 single mice from older experiments (Figure 2—figure supplement 3B). Thus we inserted the new Figure 2—figure supplement 3C.

1.3) The audience of this manuscript will be far more familiar with traditional methods of FACS analysis than PRI. As a consequence, in order to establish the robustness of PRI-based quantification and familiarity with the approach, Figure 3A should include also samples from young mice.

We included conventional plots of young mice into Figure 3A and B.

It seems from the analysis of the flow cytometry plots of old mice (Figure 3A) that all IL-21^+^ cells co-express IL-2, TNF, and IFN while being CXCR5^-^, PD-1int, and IL-10^-^. However, PRI assessment of the same samples (Figure 3F) provides a visual representation suggesting that the vast majority of IFNγ^+^ cells are also producing IL-21, while in the dot plots (Figure 3A) only ~1/7 of IFNγ^+^ cells produce IL-21.

We admit that too many frequencies given in the old Figure 3F were confusing. Therefore, we depicted here (Author response image 1) and in the novel Figure 3F and G only the most important green numbers (Z^+^ producers of all CD44^+^ cells in the respective quadrants).

The old figures showed that 5.7% (Author response image 1 left picture) and 6.8% (Author response image 1 right picture) of CD44^+^ cells are IL21^+^IFN-γ^+^ (DP) cells and 5.4% are triple-producers (with PD-1^+^). Most of the IFN-γ producing cells (32%) were IL-21^-^ and PD-1^+^.

Furthermore, looking at the PRI of young mice it is unclear whether the IL-21 producing cells maintain the same phenotype as the cells from old mice, like the authors claim (subsection “Bin plots revealed an IL-21^+^ super-functional T cell subpopulation”). For all of the above Figure 3A should allow independent assessment of young/old phenotype in addition to PRI plots of Figure 3F.

We included plots of young mice into Figure 3A and G.

The frequencies of IL-21^+^ and IL21^+^IFN-γ^+^ among the Tmem cells (CD44^+^) are not significantly different between young and old mice but show a slight tendency to be a bit higher in old diseased mice as shown both with PRI-data (Figure 3—figure supplement 1B) and FlowJo-data (Figure 3C). However, the properties of these cells to co-produce PD-1 increased significantly (Figure 3—figure supplement 1C).

1.4) Additionally, the most biologically relevant statistical assessment is in respect to cell populations (defined to the best of our ability) rather than individual markers. If the authors claim the importance of a novel cell subset defined as IL-21^+^, IFNγ^+^… the relevant statistical analysis of Figure 3E should be based on the frequency of that cell subset in young vs. old mice rather than (or in addition to) the presented analysis based on individual markers.

In fact, this consideration is really logical. Therefore, we provided two new experiments (with 10 mice in total) and included all relevant markers within one staining panel. This allowed us to analyse in parallel the subsets Tfh, Tph, and Tsh. The new Figure 3F, G, and Figure 3—figure supplement 1B show the PRI data of concatenated measured cells (3 mice each for young and old mice). The conventional flow cytometric analysis is shown in Figure 3C.

1.5) The phenotype of IL-21-producing cells displayed in Figure 4 is not easy to interpret given the poor identification of the IL21^+^ cells, as they are contained in bins together with other cells.

We agree to this. However, we experienced that taking IFN-γ instead of IL-21 is better for visualisation of preferred expression areas of the most Z-parameters. The range of IFN-γ is much greater than the range from IL-21. Therefore, tendencies and differences in the populations are better visualised as can be seen e.g. for CD40L intensities (Figure 4—figure supplement 1A) which correlates with the IFN-γ intensities.

Lack of statistics is also a concern. As a key objective of the manuscript is to show how PRI compares with FACS analysis, the figure would improve with traditional dot plots of gated IL-21^+^ cells (or even how IL21^+^IFNγ^++^ compared with IL-21^-^IFN^++^ for the different markers – to a large extent these two cell populations are in the same bin region). Statistics could then be provided and compared with PRI data.

The new Figure 3C and the new Figure 3—figure supplement 1B, offer the possibility of a direct comparison of the statistical data obtained by conventional gating and PRI analysis, respectively.

1.6) It is also puzzling why the authors do not compare the phenotype of IL-21^+^ cells and IL-21^-^IFN^+^ and IL21^-^IFN^-^ instead of a complex matrix of comparisons based on IFNγ and PD-1 expression (Figures 5, 6 and Figure 6—figure supplement 1). A major goal of the manuscript is to define a IL-21^+^IFNγ^+^ cell population that can be easily gated. Data should be provided for the population of interest in addition to arbitrary quadrants.

The results from subsequent gating are shown in the new Figure 3C.

1.7) Figure 2C. How was the determination of PD-1^-^, PD-1^low^ and PD-1 positivity determined? The gating does not follow the cell density plots.

For PD-1^+/-^ gating we used the FMO controls. For determination of PD-1^high^, we took the decline in co-expression of PD-1^high^ and IL-2 intensities in old diseased mice.

1.8) The authors note the data are reproducible. but how was reproducibility shown statistically?

Plotting the same data file produces the same pattern and numbers (100% identical) if all parameters, such as the defined threshold value of the parameter Z, the minimum number of cells per bin and the bin width are maintained. By applying different threshold values for parameter Z (IL-10) as shown in Figure 2—figure supplement 3D, the reviewer can get an impression about the changes of the pattern.

2) Biological issues2.1) The authors focused only on the "super-functional Tfh-like cells", even though in the sicker mice there is not only and expansion of IL21^+^ IFNγ^+^ PD1^low^ cells but also of IL21^+^ IFNγ^+^ PD1^high^ cells (Figure 3E) as well IL10^+^ IFNγ^+^ PD1^high^ cells.

Yes, the reviewer is right (seen also in the new Figure 3C), there are also more IL21^+^ IFNγ^+^ PD1^high^ and IL10^+^ IFNγ^+^ PD1^high^ cells in old mice compared to young mice. However, the majority of IL-21^+^ cells are PD-1^low^ and IFNγ^+^ , which defines a previously unknown subset. Therefore, we concentrated on characterisation of this novel cell subset.

To prove that this IL21^+^ INFγ^+^ PD1^low^ CD40L^high^ CXCR5^-^ Bcl6^-^ T cell population helps B cells though IL21 and CD40L, they sort non Tfh-Th1 like cells (CD4^+^, CXCR5^-^, CD44^hi^, CXCR3^+^ cells) and check their ability to support B cell activation and Ig production. Two main concerns arise from these set of experiments. First, the sorting strategy should include PD1 in order to enrich the non Tfh-Th1 like mem T cells (CD4^+^, CXCR5^-^, CD44^hi^, CXCR3^+^) within the PD1^low^ population (CD4^+^, PD1^low^, CXCR5^-^, CD44^hi^, CXCR3^+^ cells). Second, the B cell helper assay should be repeated and compared with other cell populations, including conventional Tfh1 cells (CD4^+^, PD1^hi^, CXCR5^+^, CD44^hi^, CXCR3^+^ cells) and include naïve versus total B cells.

To address this concern, we conducted a new set of T/B co-culture assays with cells sorted according to their PD-1 expression.

We performed new T/B co-culture assays to compare directly the B helper abilities of Tsh cells defined as CD4^+^, **PD1^low^** CXCR5^-^ CD44^+^ CXCR3^+^ cells, to that of CD4^+^**PD1^low^** CXCR5^-^ CD44^+^ CXCR3^-^ cells and PD-1^high^ cells, defined as CD4^+^**PD1^high^** CXCR5^+/-^ CD44^+^ containing both Tfh and Tph cells. These experiments are reported in Figure 7C and Figure 7—figure supplement 2. Due to the rarity of Tfh and Tph populations and the limited availability of old enough NZBxW F1 mice, we had to resort to aged (15-24 months) C57Bl/6 mice known to have increased autoimmune features (see for example Nusser et al., 2014). We had found that 24-month old C57Bl/6 mice presented with increased numbers of CD44^+^ cells and Tsh with the same features as sick NZBxW F1 mice. The strategy to sort the compared populations, as well as their respective productions of CD40L, IL-21 and IFN-g are presented in Figure 7—figure supplement 2A and C.

The limiting numbers of PD-1^hi^ cells restricted our study to the use of total B cells defined as CD19^+^ B220^+^ cells (Figure 7—figure supplement 2B).

These experiments clearly show dichotomic roles for Tsh and PD-1^high^ cells. While Tsh cells promote higher B cells expansion and GC-like differentiation of the B cells (GL7^+^ Fas^+^), PD-1^high^ cells promote cell survival and sustains better plasma cell differentiation (Figure 7C and Figure 7—figure supplement 2D). These results showing the superiority of Tsh cells in providing help for B cell proliferation is now described and discussed in the manuscript.

In addition, the authors should perform a more thorough characterization of the main three expanded cell subpopulations by RNAseq to better define the uniqueness of the novel "super-functional" cells.

We agree that RNAseq data might reveal additional unique markers of this novel subset. Here, we rather concentrated on comprehensive protein pattern (in particular cytokine pattern) characterisation. We feel that this analysis already provided unique markers not only for definition of this subset but also unique functional properties.

2.2) The authors claim IFNγ^+^IL-21-producing cells represent a novel population distinct from previous defined populations with "super-functional" properties.2.2.1) One cell population that has been shown important in human SLE and bears some similarities was not discussed: T peripheral helper (Tph) cells, first described in rheumatoid arthritis by Rao and colleagues (Rao et al., 2017), and this year in SLE (Bocharnikov et al., 2019), in celiac disease (Christophersen et al., 2019) and T1DM (Ekman et al., 2019). These cells are CXCR5^-^ Bcl6^-^ and characterized by IL-21 production, a critical difference to the population discussed in this manuscript is that Tph are PD-1^hi^. However, if the two populations are different, they should both be present. Do the authors claim the only T cell population producing IL-21 is the newly discovered? Is there an explanation for poor identification of other IL-21-producing T cells among CXCR5^-^ cells, namely Tph?

The reviewer is right, both cell subsets Tph and Tsh are increased in aged NZBW mice and produce IL-21. To analyse Tph cells in parallel to Tfh and Tsh cells, we performed 2 new experiments which included staining for CXCR5 (seen in Figure 3). In addition we re-analysed data of Figure 5 to directly compare the frequencies of both IL-21-producing subsets. Tsh cells are more than twice the number as Tph cells in the NZBW model.

2.2.2) Considering that the super-functional T cell population identified by the authors is similar to Tph cells, it would be of interest to know if the super-functional T cells are Blimp1^+^, which is likely the case, recognizing the potential limitation of flow staining for this transcriptional repressor in mouse.

We tried Blimp1^+^ staining, but were not successful. An example is shown where we compared staining obtained for PD-1^low^ CXCR5^-^ IFN-g^+^ IL-21^+^ (Tsh) cells, PD-1^high^ CXCR5^+^ (Tfh) cells and PD-1^high^ CXCR5^-^ cells. Compared to FMO staining, the antibody gave a similar signal for the entire population, including supposedly negative Tfh-cells, whereas T-bet staining as control gave a differential signal.

**Author response image 2. respfig2:** 

2.2.3) While the T-B cell helper assay adds a functional component to the paper, how physiological this experiment really is? Do the authors think that their T cells drive "naïve" B cells to produce Ig in vivo? Do super-functional T cells come into contact with naïve B cells in secondary lymphoid organs or elsewhere? Wouldn't naïve B cells need an IgM as well as TLR and/or cytokine signals first, to move into proximity with activated T cells?

We agree with the reviewer. However, we would like to point out that the coculture experiments have now been repeated with total B cells and that it is quite impressive that Tsh cells can help naive B cells.

2.2.4) The functional assay cannot be easily interpreted with the available information. Since sorting was done with an indirect strategy (based on CXCR3^+^CXCR5^-^ T cells) it will be important to show what is the frequency of the IL-21^+^IFNγ^+^PD-1int cells among the sorted population. A positive control with bona fide CXCR5^+^PD1^hi^ Tfh cells should be provided to compare the potency of Tfh and the claimed "super-functional" population. Titrations may be provided to assess relative potency, if needed. It is hard to claim super-functional properties without a functional comparison with the current standard of B cell help (i.e. Tfh cells). FACS plots with B cells stained with anti-GL7 and anti-IgG1 could also make the results more compelling, given the low levels of IgG antibodies produced.

A new set of co-culture assays with a sorting strategy based on the expression of PD-1 and CXCR3 (Figure 7C and Figure 7—figure supplement 2) addresses the reviewer’s concern. This new test confirmed the high helper functions of PD-1^low^ CXCR3^+^. Moreover, we characterized the respective cytokines and CD40L productions of the sorted CD4 T cell subsets to allow a direct correlation between their profiles and helper capacity (Figure 7—figure supplement 2C). These experiments provide a solid proof that Tsh cells have superior helper ability to support B cell proliferation over Tfh, suggesting a role in the production of self-reactive B cells in autoimmune contexts.

We took the advice of the reviewer and analysed the co-cultured B cells at the endpoint of the We took the advice of the reviewer and analyzed the co-cultured B cells at the endpoint of the experiment (day 5) by flow cytometry, using GL7 and Fas as marker of activated GC-like cells, and CD138 as an alternative to IgG1 for definition of antibody-producing cells. We preferred CD138 over IgG1 because IgG1 is mainly produced in Th2 contexts and Tph display a more Th1 phenotype with IFN-g, which induces diverse Ig class switching. Measuring IgG1 may have led to underestimate the production of Ig-producing cells. This new readout highlighted differences in the quality of T cell help provided by the different subsets, but also provided an explanation for the low levels of IgG detected by ELISA, given the low percentage of CD138^+^ cells.

2.2.5) Frequently, BCR agonists (anti-IgM for instance) are provided in addition to T cell stimulation for in vitro assays (see for instance Sage, Nature Immunol 2016 17:1436), this may improve the assay read-out.

Thank you for the advice. However, because we worked with limiting numbers of T cells, we could not endeavour to set an experimental system in which T help and BCR agonists are mutually titrated to optimize the co-culture assay. We plan to explore Tsh helper functions in antigen-dependent contexts involving BCR engagement in the future.

2.2.6) More clarity about disease definition would be helpful. "Chronic inflammation, high proteinuria", is not so specific, not is age of mice. Likewise, were autoantibodies and renal histology assessed?

Disease definition concerning age, proteinuria, and auto-antibodies is depicted in Figure 2A and in the main text. Auto-antibodies increase with age and disease score. Renal histology was not assessed.